# The RNA-binding protein landscapes differ between mammalian organs and cultured cells

Joel I. Perez-Perri [1], Dunja Ferring-Appel[1], Ina Huppertz[1,3], Thomas Schwarzl [1], Sudeep Sahadevan[1], Frank Stein [1], Mandy Rettel [1], Bruno Galy [2,4] ✉ & Matthias W. Hentze [1,4] ✉

System-wide approaches have unveiled an unexpected breadth of the RNA-bound proteomes of cultured cells. Corresponding information regarding RNA-binding proteins (RBPs) of mammalian organs is still missing, largely due to technical challenges. Here, we describe ex vivo enhanced RNA interactome capture (eRIC) to characterize the RNA-bound proteomes of three different mouse organs. The resulting organ atlases encompass more than 1300 RBPs active in brain, kidney or liver. Nearly a quarter (291) of these had formerly not been identified in cultured cells, with more than 100 being metabolic enzymes. Remarkably, RBP activity differs between organs independent of RBP abundance, suggesting organ-specific levels of control. Similarly, we identify systematic differences in RNA binding between animal organs and cultured cells. The pervasive RNA binding of enzymes of intermediary metabolism in organs points to tightly knit connections between gene expression and metabolism, and displays a particular enrichment for enzymes that use nucleotide cofactors. We describe a generically applicable refinement of the eRIC technology and provide an instructive resource of RBPs active in intact mammalian organs, including the brain.

RNA-binding proteins (RBPs) constitute a versatile ensemble of proteins that play key roles in fundamental biological processes. They orchestrate the life cycle of messenger RNAs, from their synthesis in the nucleus to their translation and decay in the cytoplasm, and are thus essential for shaping cellular proteomes. They are also essential for the processing, function, and decay of all other classes of RNA. There is growing evidence that protein activity can conversely be regulated by RNA[1]. Illustrating the importance of RBPs for cellular homeostasis, numerous diseases, including neurological disorders and cancer, have been linked to RBP malfunction[2].

Unbiased, system-wide approaches have paved the way for the determination of the composition, subcellular distribution, and

dynamics of RNA-bound proteomes[3–8]. These methods start with the crosslinking of RBPs to RNA in cellulo to stabilize RNA–protein interactions that occur within the native cellular environment. Irradiation of cells with ultraviolet (UV) light has been widely used to crosslink single-stranded nucleic acids and proteins with high specificity, because UV light is inefficient in protein–protein crosslinking; hence it selects for direct RNA–protein interactions. The crosslinked RNA–protein complexes are subsequently isolated under highly stringent, denaturing conditions. In RNA interactome capture (RIC), oligo(dT)-coated magnetic beads are used to select polyadenylated transcripts together with their crosslinked RBP partners[3,4]. In an enhanced version called eRIC, the capture probe is modified with locked nucleic acids (LNA),

[1]European Molecular Biology Laboratory, Meyerhofstrasse 1, 69117 Heidelberg, Germany. [2]German Cancer Research Center (DKFZ), Division of Virus-associated Carcinogenesis, Im Neuenheimer Feld 280, 69120 Heidelberg, Germany. [3]Present address: Max Planck Institute for Biology of Ageing, Joseph-Stelzmann-Str. 9b, 50931 Cologne, Germany. [4]These authors jointly supervised this work: Bruno Galy, Matthias W. Hentze. ✉e-mail: b.galy@dkfz.de; hentze@embl.org

improving specificity and the signal-to-noise ratio[5,9]. Methods to isolate the whole RNA-bound proteome regardless of RNA biotype have also been developed[6–8,10,11]. After the different capture strategies, the RNA-bound polypeptides are retrieved and analyzed by mass spectrometry.

With only a few exceptions from non-mammalian model organisms such as *Drosophila* embryos[12,13], *Caenorhabditis elegans*[14], zebrafish[15], and plants[16–18], RBP profiling methods have principally been used to study the RNA-bound proteomes of unicellular organisms or cultured cells. The lack of knowledge regarding mammalian organs and tissues is owed to technical limitations. The low penetration depth of UV light into biological specimen[19] limits the applicability of UV crosslinking in large multi-cellular organisms. A first approach to characterize the RNA-bound proteome of a mammalian organ was recently reported, which identified a relatively limited set of 119 RBPs active in mouse liver[20]. This study used formaldehyde crosslinking and hence required measures to reduce contamination from protein–protein crosslinking. A method for the sensitive and specific detection of the RNA-bound proteomes of mammalian organs is thus still missing and the subject of this report.

We adapted the stringent eRIC protocol by using cryosectioning of ex vivo specimens to comprehensively characterize the poly(A) RNA-bound proteomes of the brain, liver, and kidneys from the house mouse *Mus musculus*. We also adapted the eRIC methodology to isolate and characterize the non-poly(A) RNA-bound proteomes of these organs. Our work represents an in-depth profiling of RBPs from mammalian organs, refining the scope of RBPs, and revealing remarkable differences in RBP activity between organs as well as between organs and cultured cells.

## Results

### Refinement of eRIC to characterize the poly(A) RNA-bound proteomes of mammalian organs

The poor penetration of UV light through biological material[19] restricts UV crosslinking of RNA–protein contacts to the surface of intact tissues. To overcome this limitation and to determine the RNA-bound proteomes of murine liver, brain, and kidneys, we flash froze intact dissected organs in liquid nitrogen. The frozen organs were sectioned in a cryostat and the sections were transferred onto slides placed on a metal surface in contact with dry ice (Fig. 1a). Subsequently, the slices were exposed to 1 J/cm² UV light, following titration experiments to maximize RBP recovery while preserving specificity (Supplementary Fig. 1b), and scraped into denaturing lysis buffer. To control for background, every other organ slice was lysed without prior exposure to UV. Subsequently, eRIC was applied, starting with the isolation of poly(A) RNAs with magnetic beads conjugated to LNA-modified oligo(dT) probes, followed by extensive washes to remove non-crosslinked proteins, RBP elution by RNase treatment, and finally RBP identification by mass spectrometry[5,9]. Approximately 10% of the washed beads were heat-eluted to analyse the captured RNA (Fig. 1a).

Capillary electrophoresis analysis of the RNA captured from the three tissues confirms the effective enrichment of mRNA, combined with a strong depletion of the highly abundant non-poly(A) RNAs that otherwise prevail, such as rRNA and tRNA (Supplementary Fig. 1a). As previously observed with cultured cells[5,21], *18S* rRNA is profoundly reduced but not eliminated. The capillary electrophoresis data were corroborated by quantitative reverse transcription PCR (qRT-PCR) analyses performed with equal amounts of RNA from eRIC eluates and inputs, showing a 50- to 100-fold enrichment of two housekeeping mRNAs (*Actb*, *Gapdh*) over *18S* rRNA (Fig. 1b). Importantly, the direct qPCR analysis of eRIC eluates without prior reverse transcription revealed that *Actb* and *Gapdh* cDNAs are at least 10–100 times more abundant than DNA from the same loci, indicating that gDNA contamination is minimal (Fig. 1c).

To conduct proteomic analysis, four irradiated eRIC samples per organ were first generated, each derived from the respective organs of a single mouse. To obtain sufficient material, we combined the eRIC eluates obtained from two mice, rendering two irradiated samples per tissue. Organ sections from four mice were pooled to generate one non-crosslinked eRIC control per organ studied. RBP peptides were tandem mass tag (TMT)-labeled and analyzed in a single liquid chromatography/tandem mass spectrometry (LC-MS/MS) run (Fig. 1a). Total proteomes were also determined from input fractions for cross-comparison.

High-confidence eRIC hits were defined as proteins significantly enriched in eRIC eluates from UV-irradiated compared to no-UV samples (fold-change (FC) >2, false discovery rate (FDR) <0.05). 622, 1345, and 1238 hits were identified, respectively, from the brain, kidney, and liver (Fig. 1d and Supplementary Data 1), with remarkable reproducibility between independent experiments (Fig. 1e). Of note, total versus RNA-bound protein samples correlate poorly, indicating specific enrichment of RBPs (Supplementary Fig. 1c).

Overall, the eRIC hits identified in the three organs are enriched for RNA-binding domains (RBD) (Fig. 2a), as expected for a comprehensive set of RBPs. A gene ontology (GO) analysis also shows that eRIC hits are associated with terms mainly related to RNA metabolism, including translation, ribonucleoprotein complex, or mRNA binding (Fig. 2b and Supplementary Data 8). Moreover, most eRIC hits belong to the RNA-related protein groups "nucleic acid-binding protein" and "translational protein" according to the PANTHER classification system[22] (Fig. 2c). These enrichments indicate that eRIC captures the core mRNA-bound proteome from organs. However, and in agreement with previous RBP profiling studies from cultured cells, the mRNA-bound proteome of mouse organs also includes many proteins that lack a direct relationship to RNA metabolism a priori, as discussed below.

Taken together, ex vivo eRIC captures poly(A) RNA and crosslinked RBPs with high specificity and reproducibility from intact mouse brain, liver, and kidneys, enabling the robust, comprehensive determination of whole organ poly(A) RNA-bound proteomes.

### Global control of RBPs in an organ-specific way

We next compared the poly(A) RNA-bound proteomes of the brain, kidney, and liver to each other. In total, 589 active RBPs are shared between the three organs, and an additional 648 active RBPs were identified both in the kidney and liver (Fig. 3a). Seventy-seven RBPs were solely detected in the kidney, and, surprisingly, only very few RBPs were exclusively identified in brain or liver. This distribution suggests that the MS analysis may have been biased toward the detection of RBPs active in the kidney. Indeed, the averaged normalized TMT reporter ion signal (signal sum) in eRIC eluates of the proteins that scored as RBP hits in at least one organ is 1.0e7, 5.3e7, and 1.4e7 in the brain, kidney, and liver, respectively (Fig. 3b, top panel and Supplementary Data 2), representing a mean eRIC intensity 5.3 and 3.8 times larger in kidney than brain and liver, respectively, and 1.4 larger in the liver compared to brain. Hence, recovery of crosslinked RBPs differs substantially across organs, with kidney > liver > brain. These marked differences in RBP binding across organs are not explained by differences in RBP expression. On the contrary: the mean abundance of these proteins is similar in the kidney and liver, and only marginally lower in the brain (Fig. 3b, middle panel). Hierarchical clustering confirms that with few exceptions (Fig. 3c, e.g. clusters 1 and 2), active RBPs are captured predominantly from the kidney, irrespective of their relative expression levels across organs (Fig. 3c, e.g. clusters 5, 7, and 10). The observed differences in RBP capture efficiency can neither be explained by disparities in the quantity (Fig. 3b, bottom panel), integrity, and/or purity of the captured RNA (Fig. 1b, c and Supplementary Fig. 1a).

**a**

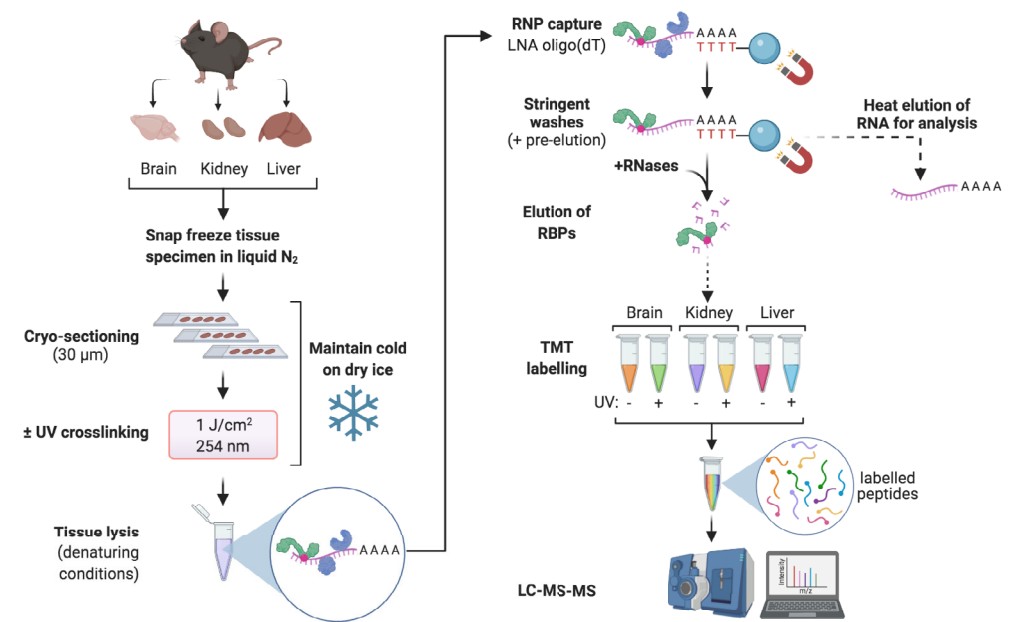

**b**

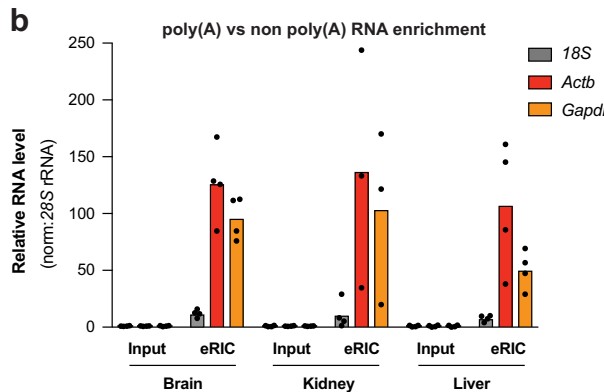

**c**

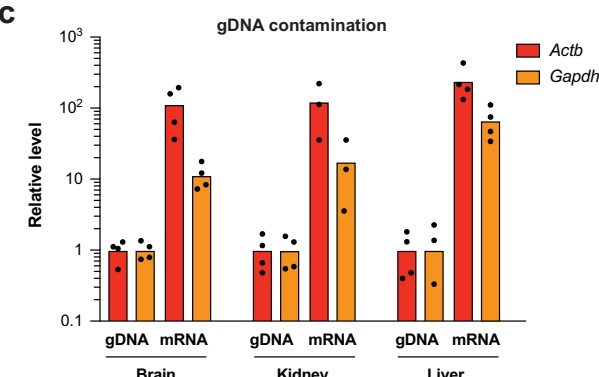

**d**

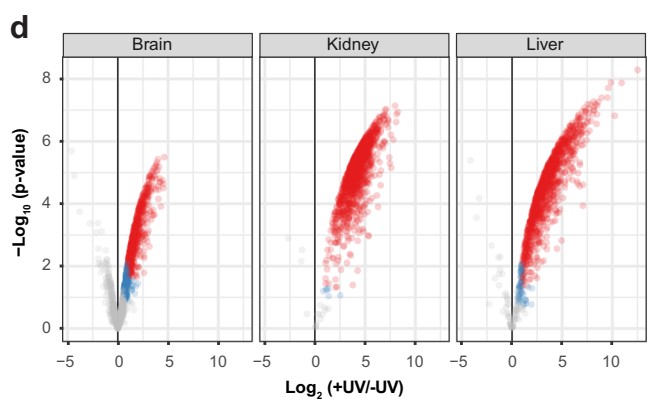

**e**

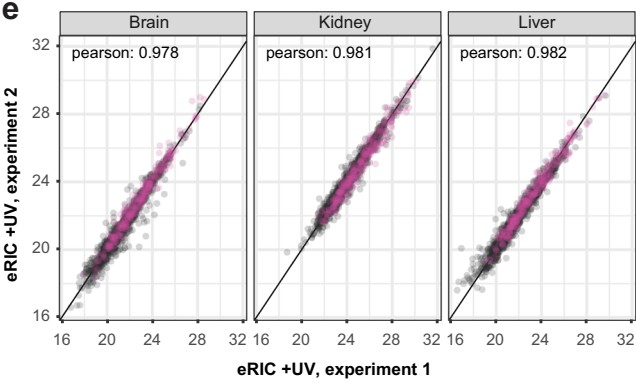

We realized that the disparities in RBP capture could result from organ-specific differences in UV crosslinking efficacy. To address this concern, we established an orthogonal, UV-independent approach to assess the poly(A) RNA-binding activity of RBPs in situ. We used a proximity ligation assay (PLA) adapted to detect protein–RNA interactions using a specific antibody against the RBP of interest and a biotinylated DNA probe complementary to the interacting RNA[23]. To identify interacting poly(A) transcripts irrespective of their sequence, we generated biotinylated oligo(dT) probes anchored to the 3'UTR/poly(A) boundaries via two randomized nucleotides at its 3'end (Supplementary Fig. 2a).

**Fig. 1 | A method for the specific determination of the poly(A) RNA-bound proteomes of mammalian organs. a** Schematic representation of ex vivo eRIC (enhanced RNA interactome capture) applied to organs. Intact flash-frozen organs are sectioned into 30 μm slices amenable for UV irradiation. Following UV cross-linking (indicated by a red dot), tissue sections are lysed under denaturing conditions. RNA-binding proteins (RBPs) bound to polyadenylated RNA are subsequently isolated under highly stringent conditions using an LNA-modified oligo(dT) probe coupled to magnetic beads[5,9]. A fraction of the isolated material is used for RNA analysis. The rest is subjected to RNase digestion to retrieve RBPs. Following solid-phase-enhanced sample preparation (SP3)[54,55], peptides subjected to tandem mass tag (TMT) labeling are multiplexed and analyzed using LC-MS/MS (liquid chromatography/tandem mass spectrometry). Created with BioRender.com. **b–e** ex vivo eRIC was used to characterize the RNA-bound proteomes of the brain, kidney, and liver from adult C57BL6/J mice. **b** RT-qPCR analysis of *18S* rRNA as well as *Actb* and *Gapdh* mRNA abundance in eRIC eluates versus input, demonstrating enrichment of mRNA. Values are expressed relative to the respective input (input mean

corresponds to 1.0). **c** qPCR analysis of mRNA versus genomic DNA (gDNA) for the housekeeping genes *Actb* and *Gapdh*, showing that gDNA contamination is minor. **b, c** n = 4 biologically independent experiments. **d** Volcano plots showing significant enrichment of RBPs in UV crosslinked over non-irradiated samples. Red dots, hits: FDR <0.05, FC >2. Blue dots, candidates: FDR <0.2, FC >1.5 (moderated two-sided *t*-test with FDR multiple testing correction). The combined ex vivo eRIC data from the brain, kidney, and liver reveal more than 1300 hit RBPs (see Supplementary Data 1). **e** Scatter plots comparing the normalized signal sums in ex vivo eRIC eluates obtained from independent experiments performed with distinct animals. Pink dots, proteins harboring known RNA-binding domains (see section "eRIC uncovers organ RBPs not previously detected in cultured cells"). **d, e** for each organ, four +UV eRIC eluates were generated, each derived from a single mouse; eRIC eluates from two mice were combined, rendering n = 2. Organ sections from four mice were pooled to generate one -UV eRIC eluate per organ (n = 1). Source data are provided as a Source Data file.

We strategically selected four RBPs that displayed higher RNA-binding activity in kidneys, although their expression levels, judged by both proteomics and immunofluorescence, are similar or even lower in kidneys than in the brain and liver (Supplementary Fig. 2b, left two panels). These include a classical RBP, the ATP-dependent RNA helicase DDX6, and three non-canonical RBPs: the glycolytic enzymes Enolase 1 (ENO1) and pyruvate kinase (PKM), and the membrane-associated amino acid transporter SLC3A2. Confirming the eRIC data (Supplementary Fig. 2b, third panel from the left), all four RBPs show stronger poly(A) RNA binding in kidneys than in the liver and brain using the PLA (Fig. 3d and Supplementary Fig. 2b, right panel). Thus, kidney RBPs appear to be highly active, and the widespread differences detected by eRIC are likely biologically determined rather than technical artifacts.

Thus, poly(A) RNA–protein interactions are globally controlled in an organ-specific manner.

## Organ-specific interactions of RBPs with poly(A) and non-poly(A) RNA

We wondered whether the overall changes in RBP activity across organs could be explained by differences in poly(A) RNA content. Poly(A) RNA levels in the brain are indeed lower than in the kidney and liver (Fig. 3e, green bars), suggesting that the globally low poly(A) RNA-binding activity in the brain could result from the relatively lower amount of poly(A) transcripts in this organ. However, while the poly(A) RNA content in kidneys is lower than in the liver (Fig. 3e, green bars), RBP binding to poly(A) RNA is higher in the kidney (Fig. 3b, top panel). Non-poly(A) RNA levels in the liver are higher than in the kidneys (and lowest in the brain) (Fig. 3e, violet bars). As cross-linking immunoprecipitation (CLIP) experiments revealed that RBPs may interact with both poly(A) and non-poly(A) RNA (ENCODE project[24,25],), we tested whether the proportion of RBPs bound to poly(A) transcripts might be affected by the stoichiometry of the two RNA biotypes. RBPs bound to non-poly(A) RNA were extracted from eRIC supernatants (depleted of poly(A) RNA) with guanidinium thiocyanate-phenol-chloroform, purified over a silica matrix, and identified by MS, Fig. 4a. The depth of this analysis was dictated by the yield in the brain, which was lower likely due to the low concentration of non-poly(A) RNA in the lysates.

Two irradiated and one non-irradiated sample were employed per organ. Non-poly(A) RBPs were defined as the proteins significantly enriched in UV-treated over non-crosslinked controls (hit: FDR <0.05, FC >2; candidate: FDR <0.2, FC >1.5). 355 non-poly(A) RBPs were identified (Fig. 4b, c and Supplementary Data 3); with 321, 322, and 328 RBPs detected in the brain, kidney, and liver, respectively (Fig. 4c). These proteins are strongly enriched in non-poly(A) RNA-related GO terms such as "structural constituent of ribosome" or "rRNA binding" (Fig. 4d), highlighting the specificity of the dataset and validating the methodology.

We then combined the non-poly(A)RIC with our previous eRIC data to determine the relative binding of proteins to poly(A) and non-poly(A) RNA (Fig. 4e and Supplementary Data 3). 222 RBPs interact with both RNA biotypes (dual binders), while 133 and 583 proteins exclusively associate with non-poly(A) or poly(A) RNA, respectively (Fig. 4f) (the latter defined as the eRIC hits that were neither detected in the aforementioned non-poly(A)RICs nor in a deeper non-poly(A) RIC described below) (Fig. 4f). Dual binders show decreased interaction with poly(A) RNA in liver relative to kidney, but this was not the case for the proteins that interact exclusively with poly(A) RNA or non-poly(A) RNA (Fig. 4g). Moreover, dual binders interact more with non-poly(A) RNA in liver relative to kidney (Fig. 4g). This is consistent with a mutually exclusive interaction of these RBPs with either RNA biotype.

Together, our results suggest that the overall poly(A) RNA content, as well as the relative levels of poly(A) versus non-poly(A) RNA may impact the composition of the poly(A) RNA interactome.

We also generated a comprehensive non-poly(A) RBP atlas of mouse liver from one non-irradiated and four irradiated samples. About 1000 proteins were significantly enriched over the non-crosslinked control (hit: FDR <0.05, FC >2; candidate: FDR <0.2, FC >1.5) (Supplementary Fig. 3a, b and Supplementary Data 4). Integration with the liver eRIC data indicates that specific sets of proteins co-purify with poly(A) and/or non-poly(A) RNA (Supplementary Fig. 3b, c). We identified 588 dual binders, and 736 or 412 proteins exclusively scoring as hits/candidates in eRIC or non-poly(A)RIC, respectively. Among dual binders, known mRNA-binding proteins (e.g., Esrp2, A1cf, Csde1, and Upf1) display a higher intensity in eRIC than in non-poly(A)RIC eluates, while the opposite is true for ribosomal proteins (e.g., Rps20, Rps8, Rpl19, and Rpl8) (Supplementary Fig. 3d and Supplementary Data 4).

A protein scores as "no-hit" if it is either not detected, or detected but not enriched over the non-crosslinked control. We classified liver RBPs as exclusive poly(A) binders if they scored as eRIC hits/candidates but were not detected in non-poly(A)RIC (n = 583) and as exclusive non-poly(A) binders if they displayed the opposite behavior (n = 404). Dual binders scored as hits/candidates in both eRIC and non-poly(A)RIC (n = 588) (Supplementary Fig. 3e). Of note, the identified poly(A) and non-poly(A) RBPs are enriched in different sets of functions, including many linked to the biology of their respective target RNAs. Terms such as "rRNA binding", "structural constituent of the ribosome", "tRNA binding", and "snRNP binding" are enriched among non-poly(A)RIC hits. Conversely, terms such as "poly(A) binding", "mRNA 3′-UTR binding" and "N6−methyladenosine−containing RNA binding" are enriched among poly(A) RBPs (Supplementary Fig. 3f). Among the 404 exclusive non-poly(A) RNA binders, 51 were ribosomal proteins (Supplementary Data 4 and Supplementary Data 9, see GO:MF "structural constituent of ribosome"). Ribosomal proteins are some of the most abundant RBPs in the cell. The finding that 51 ribosomal proteins are detected in the non-poly(A) RNP fraction but absent in the eRIC eluates shows that eRIC hits do not simply correspond to very

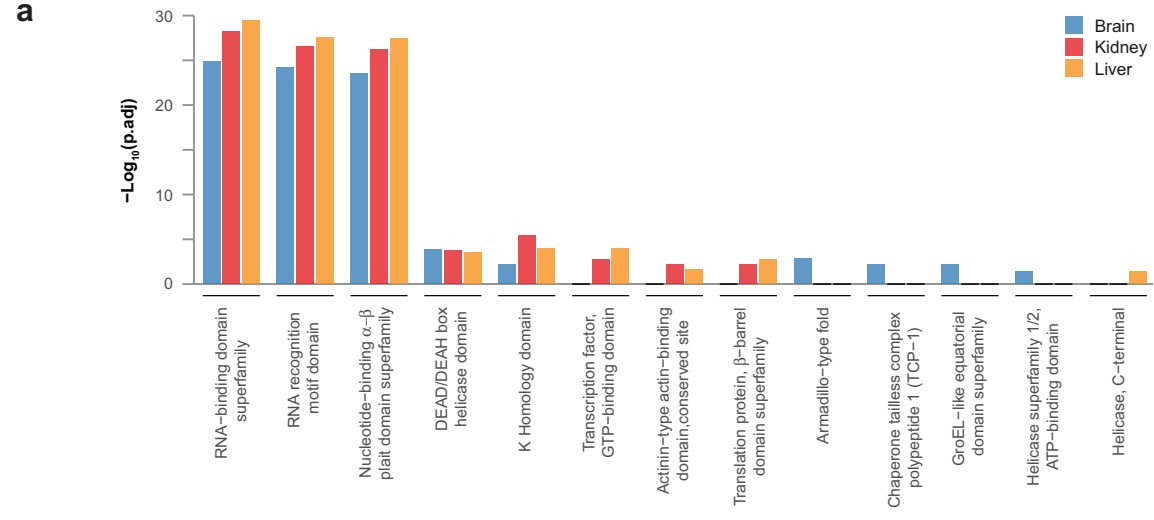

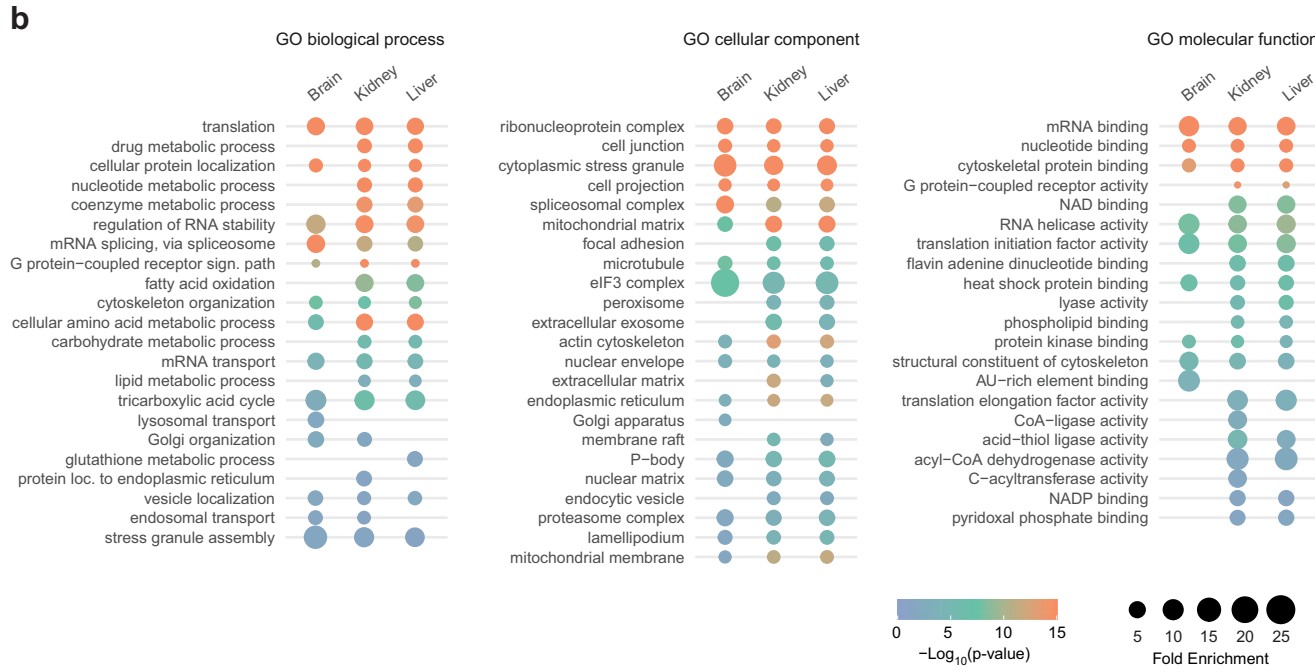

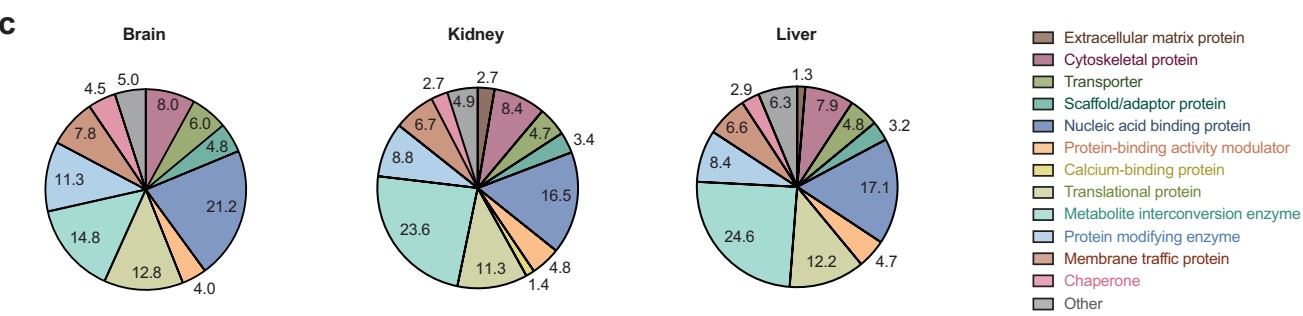

Fig. 2 | Characteristic features of the RBPs from mouse brain, kidney, and liver.
a Protein domains highly represented among ex vivo eRIC hits (Fisher's one-tailed test with independent hypothesis weighting (IHW) for multiple testing correction). Blue, brain; red, kidney; orange, liver. b Gene Ontology (GO)-term enrichment analysis (Fisher's one-tailed test with Bonferroni correction for multiple testing).

Selected GO terms corresponding to biological process, molecular function, and cellular component are displayed (see Supplementary Data 8 for the full list of GO terms). c Protein class distribution among the three tissues studied based on the PANTHER protein class ontology[22].

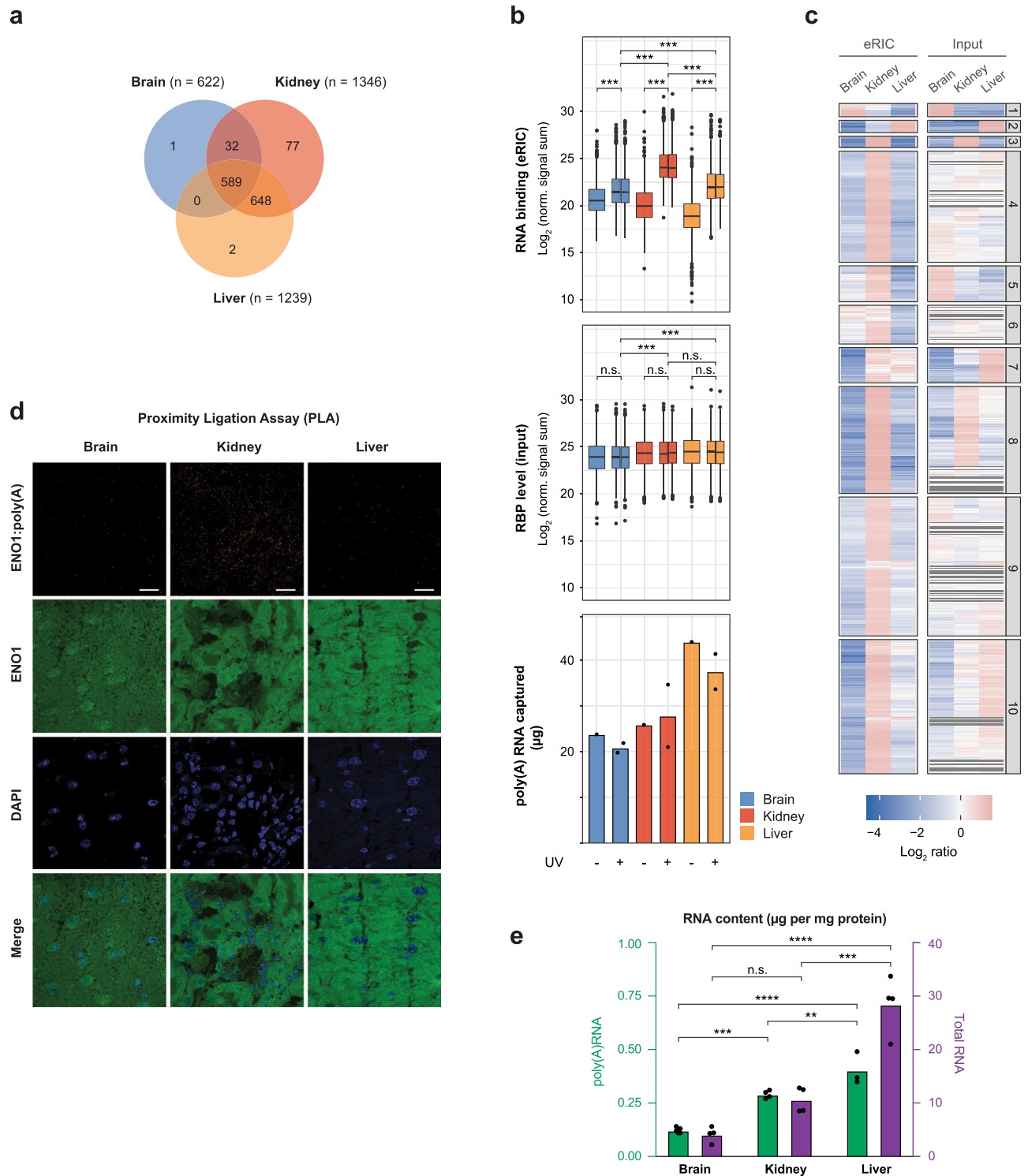

abundant proteins and provides strong additional evidence for the high specificity of our method.

**Surveying the organ-specific regulation of individual RBPs**

Our study revealed widespread differences in the overall association of proteins with poly(A) RNA across brain, kidney, and liver tissues. We next asked whether and how the binding of individual RBPs to poly(A) RNA differs between organs. For this, we analyzed the ex vivo eRIC data assuming equal mean signal intensity across samples (Supplementary Fig. 4a and Supplementary Data 5). As before, we processed proteomics data from both eRIC eluates and

total proteomic input to assess whether differences in eRIC signal intensity are due to differences in specific RNA-binding activity or RBP levels. Hierarchical clustering shows similar patterns of protein abundance in eRIC eluates and total proteomic input for the majority of RBPs (Supplementary Fig. 4b). This suggests that differential RBP binding to RNA across mouse organs most commonly results from differential RBP expression. This standard pattern has, however, numerous exceptions and dozens of RBPs exhibit differential RNA binding without commensurate changes in overall protein abundance (Supplementary Fig. 4c and Supplementary Data 5).

**Fig. 3 | Comparative analysis of RBPs from mouse brain, kidney, and liver.**
**a** Venn diagram showing the number of shared and organ-specific RBPs identified.
**b** Normalized signal sum of identified RBPs in ex vivo eRIC eluates (upper panel) and the corresponding input samples (middle panel) for each organ analyzed. Blue, brain; red, kidney; orange, liver. Center lines indicate the median, box borders represent the interquartile range (IQR), and whiskers extend to ±1.5 time the IQR; outliers are shown as black dots (pairwise comparisons using two-sided $t$-test with FDR correction, ***$p$.adj <2e-16; n.s.: not significant). Bottom panel: amount of poly(A) RNA isolated from each organ by eRIC. Note that the retrieved mass of protein (upper panel) differs extensively across the tissues analyzed (with kidney > liver > brain) and does not correlate with the mass of RNA recovered (bottom panel) (see also Supplementary Data 2). For each organ, four +UV eRIC eluates were generated, each derived from a single mouse; eRIC eluates from two mice were combined, rendering $n = 2$. Organ sections from four mice were pooled to generate one -UV eRIC eluate per organ ($n = 1$). **c** Hierarchical clustering and heatmap of the RBPs identified in the brain, kidney, and liver, showing protein abundance in eRIC

eluates (left columns) and inputs (right columns) across the three organs (shown as the Log.2 ratio of protein abundance in each eRIC or input sample relative to the average protein abundance in corresponding eRIC and input samples).
**d** Representative images of three biologically independent experiments of the proximity ligation assay (PLA) for interactions of ENO1 and poly(A) RNA, nuclear staining (DAPI), and ENO1 immunofluorescence in the brain, kidney, and liver. Scale bar, 20 μM. See quantification in Supplementary Fig. 2b. **e** poly(A) (green) and total RNA (violet) levels, in the brain, kidney, and liver, expressed as μg of RNA per mg of total protein. non-poly(A) RNA, $n = 4$ biologically independent experiments. Poly(A) RNA, $n = 3, 4, 5$ biologically independent experiments for liver, kidney, and brain, respectively. **$p$.adj < 0.01, ***$p$.adj < 0.001, ****$p$.adj < 0.0001, n.s.: not significant (one-way ANOVA with Tukey post hoc test). Poly(A) RNA: BvsK, $p$.adj = 2.76e-4; BvsL, $p$.adj = 9e-6; KvsL, $p$.adj = 9.36e-3. Total RNA: BvsK, $p$.adj = 6.34e-2; BvsL, $p$.adj = 1.0e-5; KvsL, $p$.adj = 1.24e-4. Source data are provided as a Source Data file.

These results reveal differences in the activity of individual RBPs between the brain, kidney, and liver beyond the global effects addressed above.

## eRIC uncovers organ RBPs not previously detected in cultured cells

Previous RIC studies have systematically identified hundreds of RBPs that lack recognizable RBDs or RNA-related functions (reviewed in ref. 1). In mouse organs, less than half of eRIC hits have been formerly annotated as RBPs (Fig. 5a, left panel), and no more than one-fifth bear a known RBD (Fig. 5a, second panel from the left). Consistently, many organ RBPs are associated with biological processes and molecular functions not directly linked to RNA biology (Fig. 2b, c). Overall, the intensity distributions of proteins lacking or having a discernible RBD are similar in ex vivo eRIC eluates (Fig. 1e), supporting the notion that the capture of classical and unorthodox RBPs is fundamentally alike.

We next compared the organ poly(A) RBP dataset of 1349 proteins with the large integrated atlas of published RNA-binding proteomes derived from studies conducted in mouse or human cell lines (30 datasets encompassing a total of 6518 RBPs, based on the analysis of 12 cell lines and 10 different methods for the identification of poly(A)- or total RNA-binding proteins[3–8,10,11,21,26–35]). Because we expected that the large integrated RBP atlas (6518 RBPs) includes nearly all of the 1349 organ RBPs, we were surprised to find that 291 ex vivo eRIC hits had not been found as RBPs (Fig. 5a, third panel from the left, Supplementary Data 1. Of those 291 novel RBPs, 52 were active in all three organs examined, 202 in the liver and kidneys, and 28 in the kidneys alone (Fig. 5b). These novel RBPs lack RBDs or RNA-related functions (Fig. 5a). Remarkably, more than half of these are metabolic enzymes according to the PANTHER classification system (Fig. 5c). This is in sharp contrast with the "nucleic acid-binding protein" and "translational protein" classifications that prevail within the entire organ poly(A) RBP dataset (Fig. 2c). Interestingly, 263 out of the 291 novel RBPs (>90%) are expressed in at least one of ten tested cell lines used in previous RBP profiling studies (Fig. 5d and Supplementary Data 1), with as many as 128 (~44%) expressed in at least half of them, and 42 (>14%) in all ten surveyed cell lines. Thus, the absence of the novel RBPs in previous RBP profiling studies from cultured cells does not result from a lack of expression in these cells.

As ex vivo eRIC uses a higher UV dose than previous RIC studies, we addressed whether protein–protein crosslinking contributes "piggy-back riders" as false positives to our datasets (Supplementary Fig. 5a). The proteins captured by eRIC were separated by gel electrophoresis, divided into 7 fractions of defined mass, and identified by MS to determine their distribution across the fractions. Cross-linking to other proteins should shift the signal to higher mass fractions. We applied this test in triplicate to eRIC eluates from the kidney. The fractions display a similar overall protein signal (Supplementary

Fig. 5b) but, as expected, a different composition (Supplementary Fig. 5c). The vast majority of the 306 proteins identified are present in fractions corresponding to their predicted monomeric molecular mass and thus represent bona fide RBPs (Supplementary Figs. 5d, 6 and Supplementary Data 6). This applies to both canonical (Supplementary Fig. 5e) and non-canonical RBPs as metabolic enzymes (Supplementary Fig. 5f).

Some proteins were detected in higher molecular mass fractions (Supplementary Figs. 5d, 6). Most of these have a mass close to a fraction boundary or are actually not eRIC hits (proteins with red names in Supplementary Fig. 6). A small group of proteins, including validated RBPs such as Pkm, Eno1, and Gapdh, display a bimodal distribution, with one peak at the expected mass and a second peak of high MW (Supplementary Fig. 5g). While the origin of the later peak is unknown, the presence of the former peak suggests that these proteins are genuine RBPs, in agreement with previous reports[36–38]. Only ~7.5% of the detected proteins that preferentially localize to higher MW fractions are eRIC hits and have a mass 10 KDa or less below the fraction boundary; many of these proteins are known RBPs (e.g. Sf3a1, Sfpq, Rbm14, Eef1a1, Fus, Ncl, Hnrnpu, Hnrnpul2, and Pabpn1). Differences between the observed and predicted masses could be UV-/eRIC-independent. For example, while the reported mass of Ncl is 76.7 KDa, it runs above the 100 KDa marker in both eRIC eluates and non-irradiated inputs from the liver (Supplementary Fig. 1b).

These data provide strong additional evidence for the majority of eRIC hits representing direct RNA binders rather than false positives arising from protein–protein crosslinking.

## The RNA-bound proteomes of organs lack RBPs commonly identified in cultured cells

Using the integrated RBP atlas, we noticed that 313 RBPs were commonly identified in cultured cells (≥50% of previous reports), but absent from the organ poly(A) RBP datasets (Supplementary Fig. 7a and Supplementary Data 1). Seventy-nine of these proteins were detected in the non-poly(A)RICs, leaving 221 RBPs commonly detected in cell lines that were not identified from the poly(A) or non-poly(A) RBPomes of organs. About 170 of these could not be detected in the total proteomic samples (Supplementary Fig. 7b), suggesting that they are insufficiently expressed in the tested organs. We excluded these proteins from further analysis. We thus focused on the 51 RBPs that were detected in proteomic inputs of the brain, kidney, and liver, but remained undetected by ex vivo eRIC and non-poly(A)RIC in all three organs (Supplementary Fig. 7c). Interestingly, most of these proteins correspond to bonafide RBPs associated with GO terms such as "RNA binding" and "RNA metabolic process", and >80% display nuclear localization (Supplementary Fig. 7d and Supplementary Data 9). The most highly enriched terms are linked to the biology of rRNA (e.g., rRNA processing and ribosome biogenesis) and, to a lesser extent, to

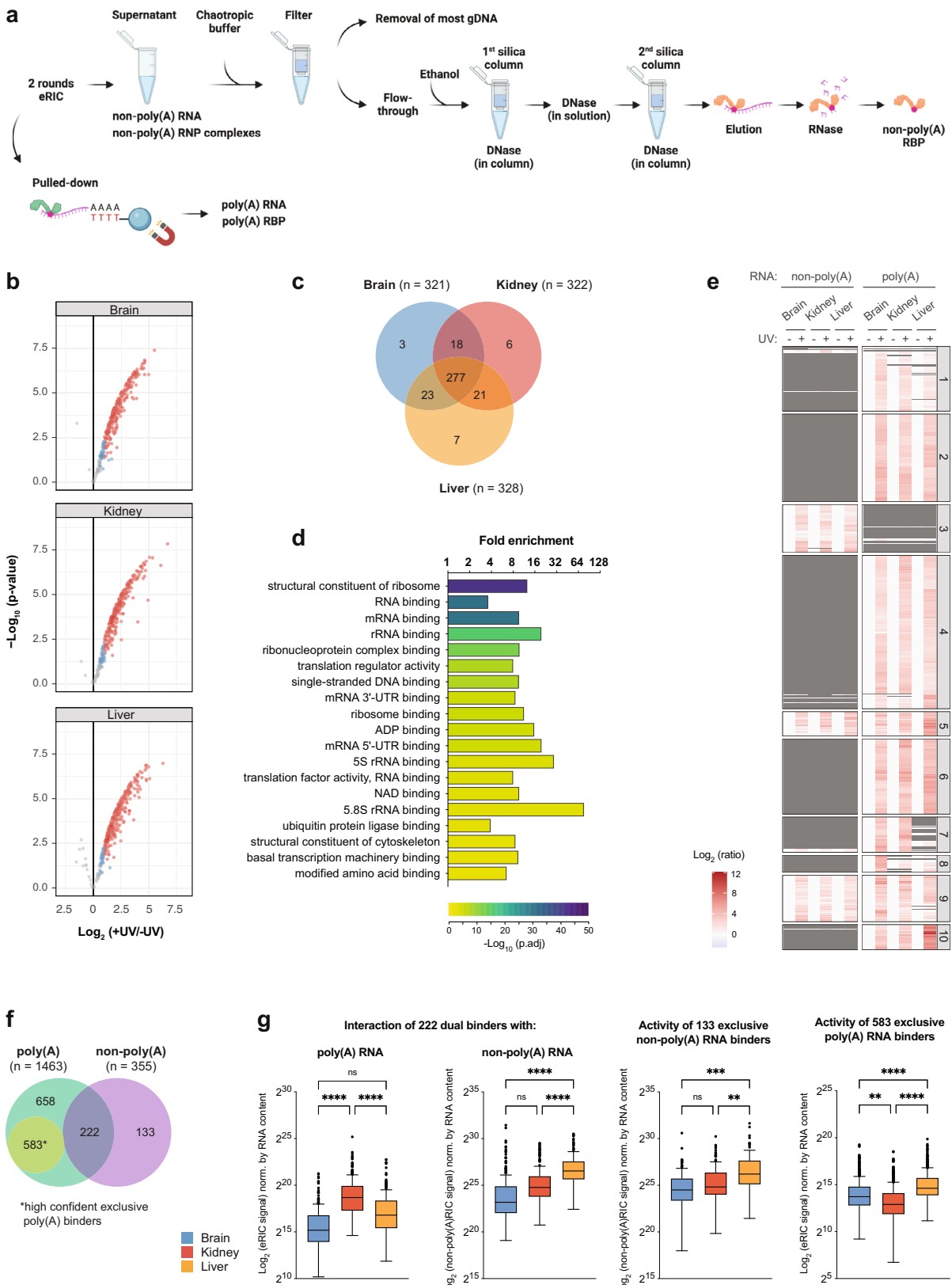

other types of non-polyadenylated RNAs (e.g., snoRNA and tRNA). A small, still significantly enriched group of proteins is involved in non-homologous end joining and telomere maintenance (Xrcc5, Xrcc6, Gnl3, Nat10, Prkdc, and Ppp1r10). Overall, this suggests that the binding of several proteins to RNA may be enhanced in cultured cells relative to organs for reasons that remain to be explored.

## Striking prevalence of metabolic enzymes as RBPs in organs

The most striking finding regarding RBPs in organs is the prevalence of enzymes. Particularly enzymes of intermediary metabolism interact with poly(A) RNA in mouse organs (Fig. 5a, fourth panel from the left, Supplementary Data 7), with at least 15% of kidney and liver RBPs and about 8% of brain RBPs being annotated as enzymes of intermediary

**Fig. 4 | Determination of the non-poly(A) RNA-bound proteomes of mammalian organs. a** Schematic representation of non-poly(A)RIC. eRIC supernatants are depleted of poly(A) RNA by oligo(dT) bead selection and hence predominantly contain non-poly(A) RNA and non-poly(A) RNP complexes that were purified by modified 2C[45]. Created with BioRender.com. **b** Volcano plots showing significant enrichment of RBPs in UV crosslinked over non-irradiated samples. Red dots, hits: FDR <0.05, FC >2. Blue dots, candidates: FDR <0.2, FC >1.5 (moderated two-sided *t*-test with FDR multiple testing correction). **c** Venn diagram showing the number of shared and organ-specific non-poly(A) RBPs identified. **d** Gene Ontology (GO)-term enrichment analysis (Fisher's one-tailed test with g:SCS multiple testing correction). Selected GO terms corresponding to molecular function are displayed (see Supplementary Data 9 for the full list of GO terms). **e** Hierarchical clustering and heatmap of the poly(A) and non-poly(A) RBPs identified in the brain, kidney, and liver, shown as the Log.2 ratio of protein abundance in irradiated versus non-irradiated samples. **f** Venn diagram depicting the number of proteins interacting exclusively with poly(A) RNA or non-poly(A) RNA or with both biotypes of RNA. High-confident poly(A) RNA binders (in yellow) are defined as the eRIC hits that were not detected neither in the non-poly(A) RNA-bound proteomes described here nor in an in-depth analysis of non-poly(A) RNA binders performed in the liver (see below). **g** Normalized signal sum of identified poly(A), non-poly(A), and dual RBPs, as appropriate, in ex vivo eRIC eluates (first and fourth panels) and non-poly(A)RIC (middle two panels). Protein signal was adjusted by RNA content (Fig. 3e). Blue, brain; red, kidney; orange, liver. Center lines indicate the median, box borders represent the interquartile range (IQR), and whiskers extend to ±1.5 times the IQR; outliers are shown as black dots. For each organ, four +UV eRIC and four +UV non-poly(A)RIC eluates were generated, each derived from a single mouse; eRIC and non-poly(A)RIC eluates from two mice were combined, rendering *n* = 2. Organ sections from four mice were pooled to generate one -UV eRIC and one non-poly(A)RIC eluate per organ (*n* = 1). **p.adj < 0.01, ***p.adj < 0.001, ****p.adj < 0.0001, n.s.: not significant (one-way ANOVA with Tukey post hoc test).

metabolism. While the systematic enrichment of metabolic enzymes as RBPs has already been noted in cultured cells (reviewed in refs. 1,39), they did not exceed 7% of the total (Fig. 5a, first panel from the right).

Enrichment analysis shows that the >250 enzymes which bind poly(A) RNA in organs are involved in over 20 pathways connected to energy, carbon, fatty acid, or amino acid metabolism (Fig. 6a, left panel). In comparison, only ten of these pathways are also over-represented in the RBP datasets from cultured cells. The fraction of enzymes of a given pathway that bind poly(A) RNA can be rather high (Fig. 6a, right panel). For example, >60% of the enzymes of the tricarboxylic acid (TCA) cycle, pyruvate metabolism, or arginine biosynthesis are ex vivo eRIC hits. Taking pyruvate metabolism (Fig. 6b), glycolysis (Supplementary Fig. 8a), and the TCA cycle (Supplementary Fig. 8b) as examples, enzyme-RBPs appear to be broadly distributed over the pathway as opposed to being limited to specific reactions.

Against this trend, only two pathways, oxidative phosphorylation and glutathione metabolism, are highly represented in cell culture but not in the organ RBP datasets (Fig. 6a). Possibly, the exposure of cultured cells to supra-physiological concentrations of oxygen and chronic oxidative stress could explain this observation, although this is speculative at present.

Interestingly, the vast majority of enzyme-RBPs also bind non-poly(A) RNA (Supplementary Fig. 3g and Supplementary Data 9). As shown in Supplementary Fig. 3g, dual binders, in fact, prevail in 21 out of the 24 metabolic pathways enriched among eRIC hits (Fig. 5a). Of note, pathways such as "metabolism of xenobiotics by cytochrome P450", "steroid hormone biosynthesis" and "retinol metabolism" are enriched exclusively among non-poly(A) RBPs (Supplementary Fig. 3g).

Taken together, the enzyme binding to RNA is even more prevalent in organs than previously observed for cultured cells. Importantly, these results indicate that this tantalizing phenomenon also occurs in living animals. It is widespread across multiple metabolic pathways, further highlighting tightly knit connections between gene expression and metabolism that await dissection in detail.

### Enzymes using nucleotide cofactors are highly enriched amongst organ RBPs

Which catalytic activities are most prevalent among the enzyme-RBPs identified in mouse organs? We selected eRIC hits that belong to the group "metabolite interconversion enzyme" according to PANTHER[22] and noticed that oxidoreductases, transferases, and hydrolases are most highly represented (Fig. 7a, top and Supplementary Data 7). While this corresponds to the overall occurrence of these enzymatic groups in the organs analyzed (Fig. 7a, bottom), the catalytic activity "ligase" is clearly enriched among enzyme-RBPs, while hydrolases are under-represented (Fig. 7b, top panel). A more refined analysis of enzyme subgroups shows an overrepresentation of two specific subtypes of oxidoreductases amongst the RBPs, namely dehydrogenases and peroxidases, while other oxidoreductase subtypes are not enriched or even slightly under-represented (Fig. 7b, bottom panel); there is also a modest enrichment for nucleotidyltransferases, and a tendency for under-representation of lipases and phosphatases (both hydrolase subtypes) amongst RBPs, as well as of kinases (transferase subtype) (Fig. 7b, bottom panel).

Domains involved in mono- or di-nucleotide binding (e.g., ATP and NAD(P)+ and FAD) have been proposed as potential interfaces for RNA binding[1,31,39]. Interestingly, the GO terms "NAD binding" and "NADP binding" are enriched among the whole repertoire of poly(A) RBPs identified in the kidney and liver (Fig. 2b, right panel) and also among dual binders (Supplementary Fig. 3f). Furthermore, nearly 30% of the identified enzyme-RBPs bear at least one classifiable nucleotide-binding domain (see Supplementary Data 10 for domain classification), which is twice as frequent as the non-RBP enzymes expressed in the same organs (Fig. 7c). Domains involved in NAD(P) (in -19% of enzyme-RBPs vs 12.6% of non-RBP enzymes) and AMP (4 vs 0.9%) binding are particularly frequent among enzyme-RBPs (Fig. 7c).

In addition to these architectural features, some enzymes may correspondingly use cofactors via domains that are still ill-defined. Therefore, we considered cofactor usage (Fig. 7d), taking into account a relatively broad range of cofactors frequently used in enzymatic reactions, including metals. This analysis confirmed that enzyme-RBPs preferentially use nucleotide cofactors compared to non-RBP enzymes, especially NAD(P) (24.8 vs 18.1%), coenzyme A (20.4 vs 9.6%), ATP (16.4 vs 12.5%), and FAD (11.2 vs 6.1%). All other cofactors, including metals, are equally used by enzyme- and non-enzyme-RBPs (Fig. 7d).

## Discussion

Since pioneering work about a decade ago, numerous analyses revealed that RNA-bound proteomes extend well beyond the previously known RBPs involved in core steps of RNA biology, at least in cultured cells and non-mammalian organisms[3–8,12–15]. The resulting atlases of RBPs have paved the way for unexpected insights into RNA biology, especially regarding the role of protein–RNA interactions in cell biological processes such as e.g., metabolism, DNA methylation, protein ubiquitination, or autophagy[37,40–43]. Nonetheless, corresponding data at the organismal level, including an assessment of organ RBPs had not been reported, likely for technical reasons.

Very recently, FAX-RIC was used to characterize the RBPs of mouse liver[20]. Because the applicability of UV crosslinking is constrained by the limited penetration depth of UV light, the authors alternatively employed 4% formaldehyde to crosslink proteins to RNA. Although formaldehyde is advantageous over UV light for crosslinking proteins to double-stranded RNA, formaldehyde, unlike UV crosslinking, efficiently promotes protein–protein crosslinks and hence necessitates measures to distinguish protein–protein crosslinking from direct RNA–protein interactions. FAX-RIC identified a total of 119

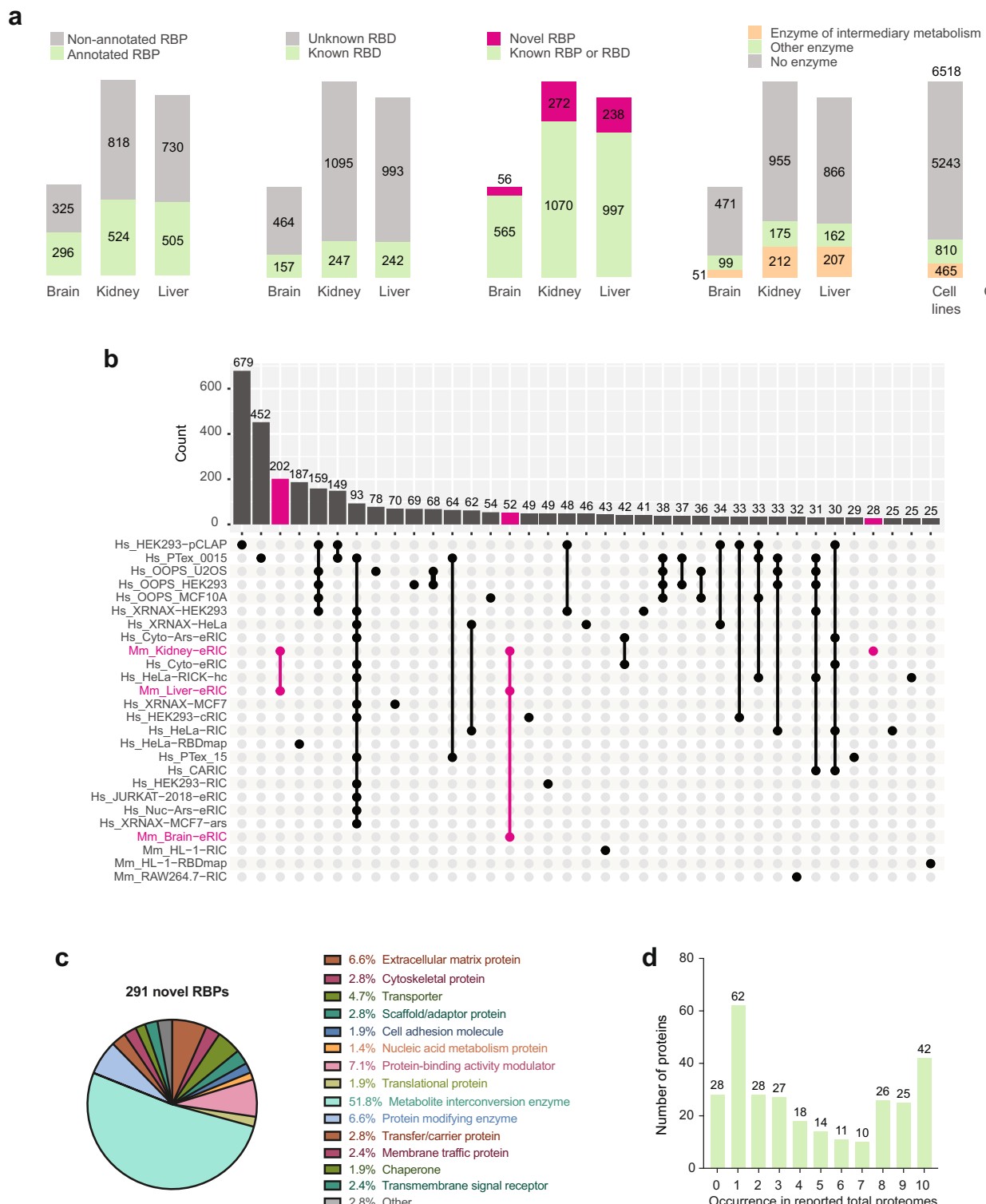

**Fig. 5 | Ex vivo eRIC uncovers many novel RBPs. a** For each organ, the number of eRIC hits that are annotated RBPs, that bear a known RNA-binding domain (RBD), and that are novel RBPs (i.e., not previously reported as RBP or as bearing an RBD, and not identified in any published study). Right: Number of RBPs identified in organs or cell lines that are (metabolic) enzymes. **b** Upset plot representing the number of RBPs (y-axis) shared between this study and published lists of mouse and human RBPs (only intersections comprising 25 or more RBPs are shown). **a**, **b** Pink, novel RBPs. **c** Protein class annotation of the novel RBPs identified in any of the three organs studied (based on PANTHER protein class ontology[22]). Note that compared to the overall RBP dataset (see Fig. 2c), the class "translational protein" is under-represented, while the category "Metabolite interconversion enzyme" is over-represented. **d** The presence of the identified novel RBPs in total proteomes of ten different cell lines employed in previous RBP profiling studies was interrogated using public data (see methods). The number of novel RBPs (y-axis) identified in inputs across an increasing number of cell lines (x-axis) is shown.

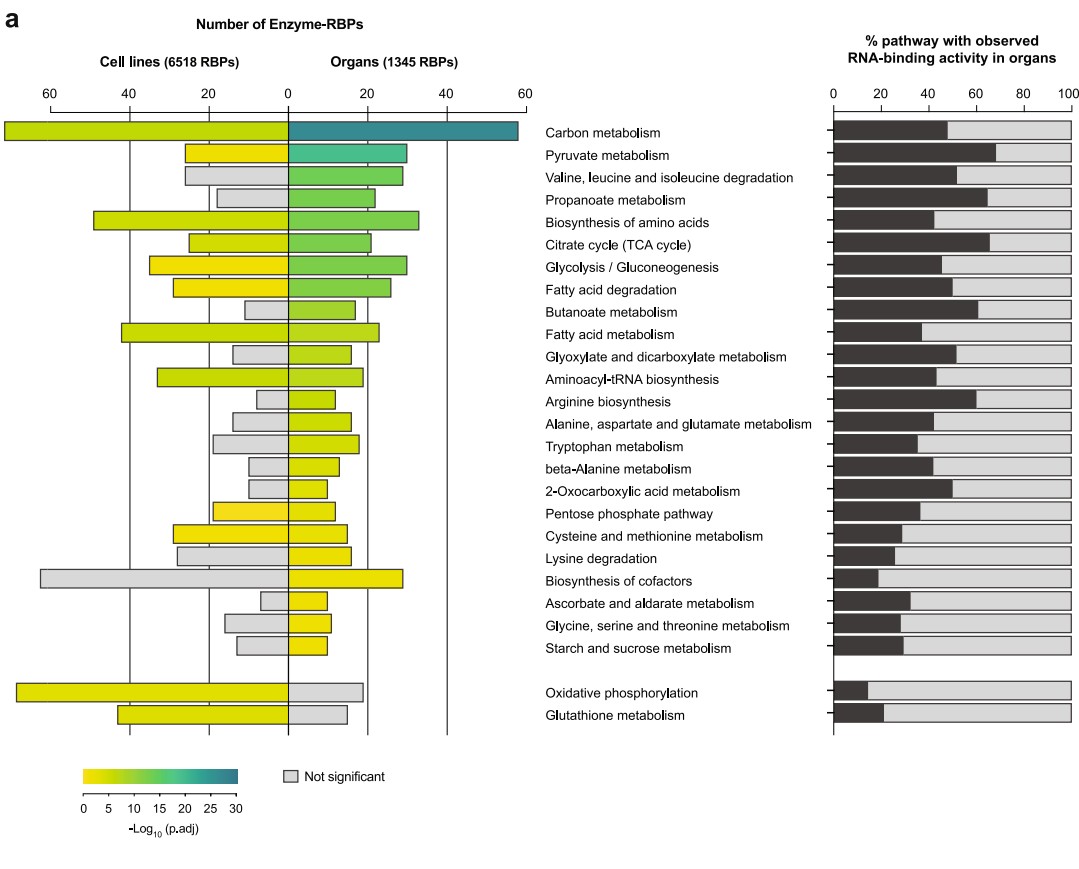

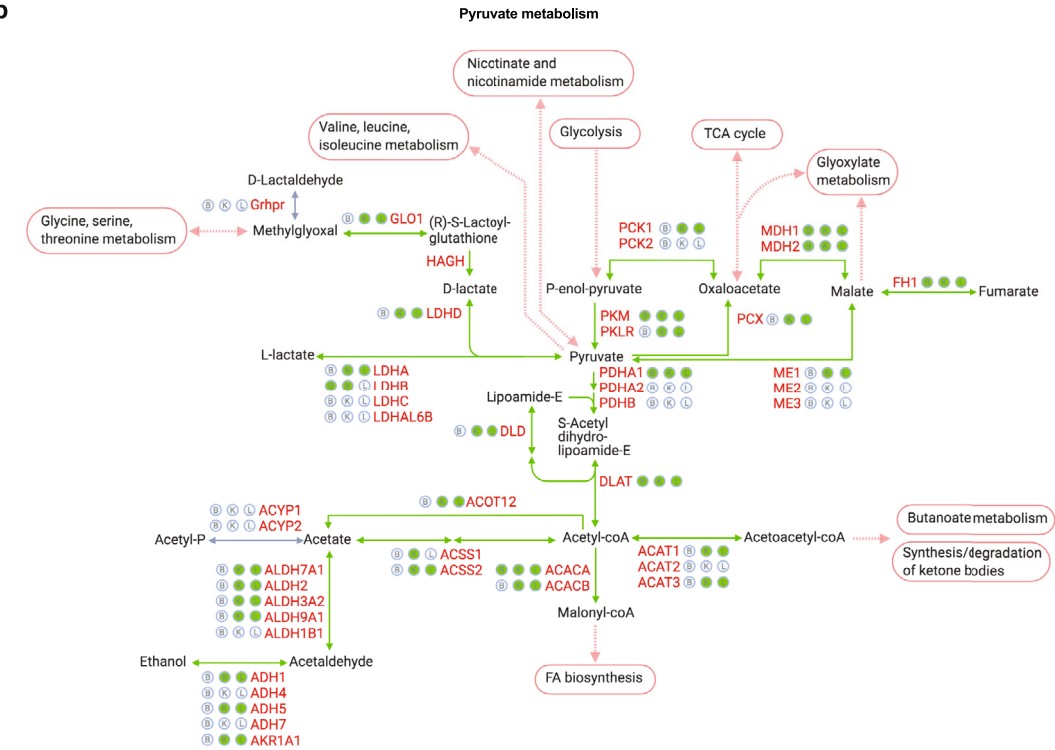

**Fig. 6 | Pervasive RNA binding of metabolic enzymes in mouse organs. a** On the left, KEGG pathway enrichment analysis among the RBPs identified in at least one organ or, for comparison, in at least one published RBP library in cultured cells (Fisher's one-tailed test with g:SCS multiple testing correction). The most enriched metabolic pathways among organ RBPs are displayed; two pathways highly represented among cell culture RBPs but not among organ RBPs are shown at the bottom. The size of the bars indicates the number of proteins identified as RBP for a given KEGG pathway. On the right, fraction of proteins in a given KEGG pathway that have been identified as RBP by ex vivo eRIC in the brain, kidney, or liver. **b** Schematic representation (based on the KEGG database) of pyruvate metabolism in mouse. Filled green circles next to protein names denote the RNA-binding activity of the corresponding enzyme in the brain (B), kidney (K), or liver (L); empty circles denote the absence of evidence for RNA association. Reactions catalyzed by enzyme-RBPs are represented as green arrows.

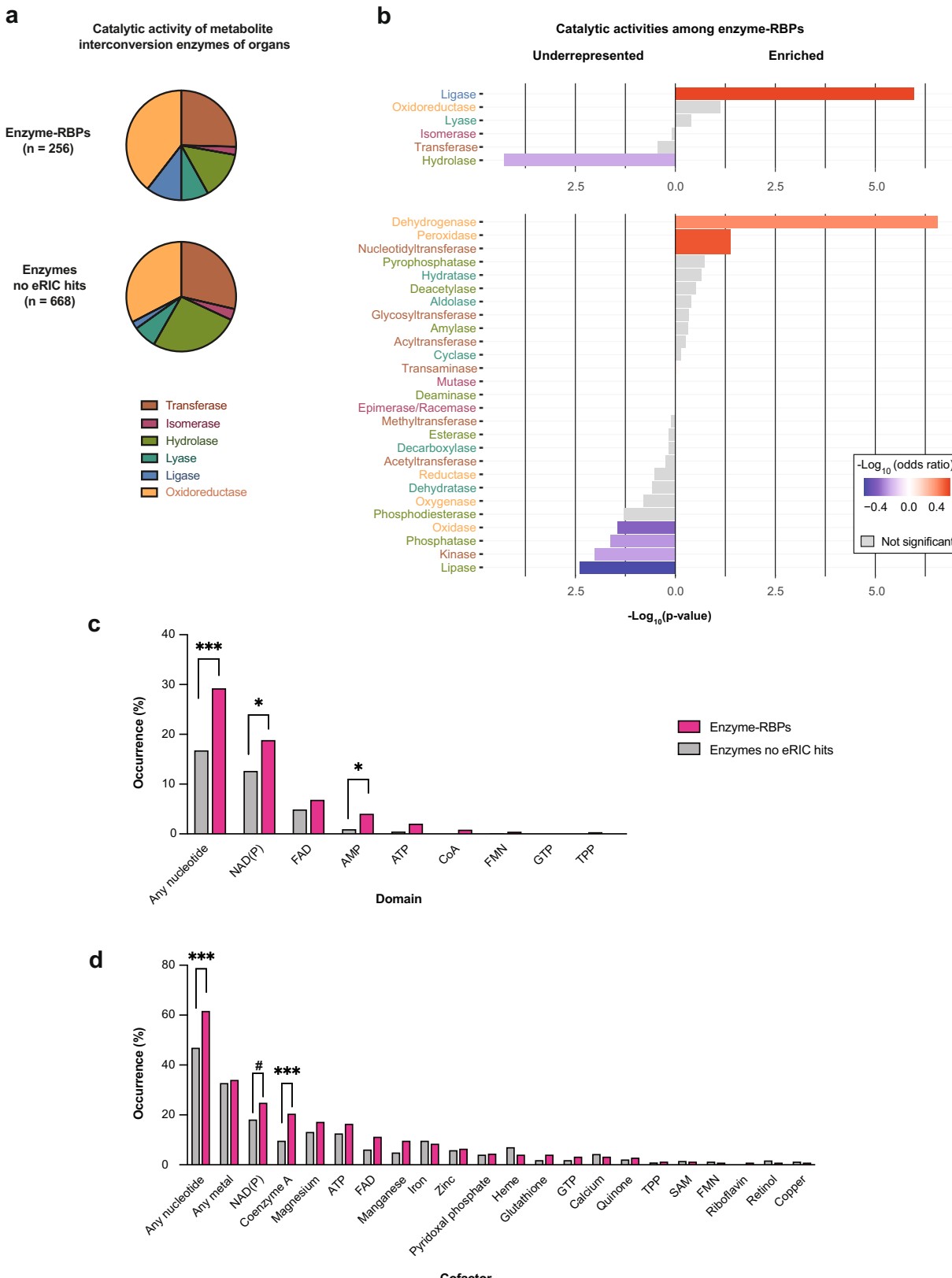

RBPs in mouse liver, far fewer than anticipated, missing numerous "housekeeping RBPs" that are part of the core mRNA-bound proteome, and reflecting the non-trivial challenges associated with the adaptation of cell-based RBP discovery technologies to mammalian organs.

Here we combined organ cryosectioning and UV crosslinking with the strengths of eRIC. Our data show that ex vivo eRIC comprehensively interrogates the RNA-bound proteomes of intact organs, uncovering both shared and organ-specific features of RBPs from mouse brain, kidney, or liver. In comparison to FAX-RIC, eRIC uncovered more than 1200 high-confidence poly(A) RBPs from the same organ (Fig. 3), resulting in comprehensive RBP atlases from intact mammalian organs.

**Fig. 7 | Enzyme-RBPs broadly interact with nucleotide cofactors. a** Enzymes of intermediary metabolism (metabolite interconversion enzymes in PANTHER protein class[22]) identified as eRIC hits in the brain, kidney, or liver (designated Enzyme-RBPs) were classified based on their catalytic activity; for comparison, the same classification was applied to enzymes of intermediary metabolism that were expressed in at least one organ but were not identified as eRIC hits. **b** Catalytic activity enrichment analysis among the enzyme-RBPs identified in mouse organs (Fisher's one-tailed test without correction for multiple comparison). The upper and bottom panels show, respectively, the main types and corresponding subtypes of enzymatic activities (indicated by the same distinct color in both panels). **c, d** Bar graphs indicating the proportion of enzymes expressed in organs and identified (pink) or not (gray) as eRIC hits that bear typical nucleotide-binding domains (any nucleotide, $p$.adj = 4.29e-4; NAD(P), $p$.adj = 4.63e-2; AMP, $p$.adj = 1.18e-2) (**c**) or employ specific cofactors (any nucleotide, $p$.adj = 8,77e-4; CoA, $p$.adj = 6.01e-4) (**d**). *$p$.adj <0.05, ***$p$.adj <0.001, #$p$.adj = 0.065 testing specifically nucleotide cofactors in (**d**) (Fisher's exact test with Benjamini–Hochberg correction).

While eRIC selects for proteins that bind poly(A) RNA, other RBP profiling methods such as OOPS, XRNAX, 2 C, or PTex have been used to identify RBPs regardless of the RNA biotype that they bind to[6–8,44]. These methods show a prevalence for proteins that interact with highly abundant, non-poly(A) RNA species such as rRNAs or tRNAs. While being more inclusive regarding the RNA biotype, they miss a significant fraction of less abundant RBPs that may exclusively or predominantly bind to the polyadenylated transcripts captured by eRIC, since poly(A) RNAs represent only a minor fraction (~3–5%) of the cellular RNA. Here we adapted 2C[44] to eRIC supernatants (depleted of polyadenylated transcripts) to purify and identify proteins associated with non-poly(A) RNA in organs. A total of 1225 non-poly(A) binders active in the brain, kidney, or liver were identified (Fig. 4, Supplementary Fig. 3, Supplementary Data 3, 4). It is highly plausible that the cryosectioning-crosslink strategy we implemented here could also be used in combination with alternative approaches for total RNA purification. Furthermore, tissue cryosectioning and UV crosslinking could, in principle, also be followed by immunoprecipitation of an RBP of interest to identify its target RNAs by CLIP-ing methods from organ samples.

Considering the enormous efforts to investigate the mammalian brain[45,46] and the essential roles of RBPs in neuronal functions[47,48], the determination of CNS-/brain-related proteome-wide RBP datasets has been overdue. Therefore, we expect the data reported here to be particularly valuable for neurobiologists. The adapted ex vivo eRIC pipeline can, in principle, be applied to many other animal models or human samples, opening novel opportunities for studying RBP biology in health and disease.

We observed unexpected and marked differences between the RNA interactomes of organs versus cultured cells. On the one hand, ex vivo eRIC identified 291 RBPs in organs not detected in any of the 30 published RBP profiles of cell lines; this is remarkable, because the cumulative organ dataset is five times smaller than that of cell lines. On the other hand, dozens of previously identified RBPs were not detected by ex vivo eRIC or non-poly(A)RIC despite being present in the tissues analyzed. Whether this reflects differences in RBP activity due to e.g., the metabolic environment (nutrient availability, oxygen levels, etc.) or the activity of oncogenic pathways in cell lines versus normal cells of healthy tissue is currently not known.

We noticed that the overall quantity of proteins crosslinked to a given quantity of poly(A) RNA differs substantially between organs, with protein binding to poly(A) RNA being globally superior in the kidney compared to the liver and brain (Fig. 3b, c). Total proteome analyses indicate these differences are not primarily due to organ-specific alterations of overall protein abundance (Fig. 3b). Similarly, the integrity and purity of the RNA recovered by eRIC are comparable across all three tissues (Fig. 1b, c and Supplementary Fig. 1a), excluding these technical artifacts. Furthermore, all ex vivo eRIC samples were analyzed in one single MS run. Organ-specific differences in UV crosslinking efficiency (due to e.g. dissimilarities in UV penetrance or the impact of UV-absorbing molecules) were also excluded as a technical reason: an orthogonal method (PLA) enabling the detection of protein–RNA complexes in situ independently of UV irradiation confirmed higher poly(A) RNA association in the kidney of four unrelated RBPs that are actually expressed at a similar or even higher level in brain or liver (Fig. 3d and Supplementary Fig. 2).

This raised the question of whether organ-specific differences in RNA content control poly(A) RNA–protein interactions. We found that poly(A) RNA levels are lowest in the brain among the organs studied, which could explain the overall reduced association of proteins to poly(A) RNA observed in the brain. Nevertheless, poly(A) RNA levels are higher in the liver than kidney, and still, the liver displays reduced poly(A) RBP activity. As many RBPs seem to interact with both poly(A) and non-poly(A) RNA (ENCODE project[24,25]), we hypothesized that the proportion of RBPs bound to poly(A) RNA might be affected by the ratio of the two RNA biotypes. In support of this notion, comparing the liver and kidney, we found that: (1) non-poly(A) RNA levels are higher (Fig. 3e), (2) dual binders but not proteins that exclusively bind to poly(A) RNA show decreased interaction with poly(A) RNA, and (3) dual binders display reduced interaction with poly(A) RNA but increased interaction with non-poly(A) RNA (Fig. 4g). Thus, the higher amounts of non-poly(A) RNA in the liver may "compete" for the binding of RBPs to poly(A) RNA.

Previous RBP profiling studies revealed that a significant fraction of the RNA-bound proteomes of cells consists of enzymes of intermediary metabolism[1,39]. Here we show that this is not a peculiarity of cultured cells. On the contrary, RNA-binding metabolic enzymes are even more prevalent in organs (Fig. 5a). While we cannot formerly exclude that some of the many individual enzyme–RNA interactions could be false positives, our observations further amplify the notion of physiologically important functions for these associations: RNA-binding enzymes have been shown to moonlight as trans-acting factors to control RNA fate, as illustrated by e.g., ACO1 or GAPDH;[36,49] conversely, direct RNA binding to enzymes has been uncovered to riboregulate their catalytic activities[37,40,50].

Our data indicate that enzymes bind to poly(A) and non-poly(A) RNA and that this binding is widespread across and within multiple metabolic pathways, and even more prominent overall in organs than in cultured cells (Figs. 5a, c, 6 and Supplementary Figs. 3, 8). Yet, compared to other metabolic pathways, enzymes involved in glutathione metabolism and oxidative phosphorylation appear to be more prone to RNA binding in cultured cells than in organs (Fig. 6a). As glutathione is a major antioxidant, the overrepresentation of the glutathione metabolism pathway in RBP profiling studies of cultured cells might reflect the supra-physiological oxygen tension in cell culture and resulting chronic oxidative stress[51]. Similarly, the overrepresentation of enzymes involved in oxidative phosphorylation may reflect differences in the activity of the electron transport chain. Performing eRIC studies in cells/organs subjected to either acute or chronic changes in e.g. oxygen levels, energy supply/demand may help address these points.

A large fraction (60%) of the enzyme-RBPs identified in this study bind mono- or di-nucleotide cofactors, with "NAD(P) binding" being particularly frequent among the kidney and liver RBP repertoires (Fig. 7c, d). Furthermore, the detected enzyme-RBPs are highly enriched for specific enzymatic functions that rely on the use of nucleotide cofactors: ligases, which employ ATP or a similar energy donor, dehydrogenases that typically use NAD or NADP as proton acceptors, or CoA that we identify as a cofactor enriched in RBPs (Fig. 7b). As ATP,

NAD, FAD, and CoA on the one hand and RNA on the other share the AMP handle as a structural feature, nucleotide-binding domains and RNA-binding interfaces may be evolutionarily connected[1,52].

Taken together, we report comprehensive atlases of RNA-binding proteins in mammalian organs. Far from only confirming the expected, our data reveal numerous surprising differences that form the basis for further explorations of the scope of RNA biology in mammalian physiology and disease. The method(s) described here should be broadly applicable to other organs and organisms.

## Methods

### Coupling of capture probes to beads

See ref. 9 for a detailed step-by-step protocol. The capture probe (HPLC purified; Exiqon) contains a primary amine at the 5′ end, a flexible C6 linker, and 20 thymidine nucleotides in which every other base is an LNA: /5AmMC6/+TT + TT + TT + TT + TT + TT + TT + TT + TT + TT (+T: LNA thymidine, T: DNA thymidine). Prior coupling, carboxylated magnetic beads (50 mg/mL; Perkin Elmer, M-PVA C11) were washed three times with five volumes of 50 mM 2-(N-morpholino) ethanesulfonic acid (MES; Carl Roth, 4256.5) buffer, pH 6.0. The washed beads were then combined with a mix made of one volume of probe solution (100 µM in nuclease-free water, Ambion) and five volumes of freshly prepared N-(3-dimethylaminopropyl)-N′-ethylcarbodiimide hydrochloride (EDC-HCl; Sigma-Aldrich, E7750) solution (20 mg/mL in MES buffer).

The coupling reaction was performed at 50 °C for 5 h with constant agitation (800 rpm in a Thermomixer). The beads were then washed twice in phosphate-buffered saline (PBS) and incubated for 1 h at 37 °C in 200 mM ethanolamine pH 8.5 (with constant agitation at 800 rpm) to inactivate residual carboxyl groups. The coupled beads were finally washed three times with 1 M NaCl and stored in 0.1% PBS–Tween at 4 °C until use.

### Mouse husbandry and organ dissection

Male mice on a homogenous C57BL6/J genetic background were housed under specific pathogen-free and light- (12:12 h light:dark cycles), temperature- (21 °C), and humidity (50–60% relative humidity)-controlled conditions. Food (Teklad, 2018S) and water were available ad libitum. The mice were sacrificed at 11 to 13 weeks of age by cervical dislocation; they were fastened 2 h prior to sacrifice (with access to water only). Animal handling (license 22-008_HD_LAR) was in accordance with guidelines approved by the animal care and use committee of the European Molecular Biology Laboratory (EMBL). Organs were immediately harvested and flash-frozen in liquid nitrogen. They were stored at −80 °C until use.

### eRIC: Cryosectioning, UV irradiation, and cell lysis

Organ sections (thickness 30 µm) were prepared in a Cryostat (Leica Biosystems, Leica CM3050 S) set to −20 °C and deposited onto SuperFrost glass slides (Carl Roth, H880); the glass slides were pre-cooled to −20 °C in the cryostat chamber to maintain the samples at the lowest temperature possible. 15–20, 20–30, and 10–15 sections from the brain, kidney, and liver, respectively, were placed on a glass slide, and a total of around 15 slides were prepared per organ per mouse (corresponding to ~80% of the brain, ~30% of the left lateral lobe of the liver, and 80% of both kidneys). Each section contains the following approximate quantities of poly(A) RNA and total protein (numbers extrapolated from the amount of cryosections used and the quantity of RNA and protein present in eRIC eluates and pooled samples, respectively): brain: 41.6 ng of poly(A) RNA and 342 µg of protein; kidney: 35.5 ng of poly(A) RNA and 94.4 µg of protein; liver: 113.4 ng of poly(A) RNA and 304.2 µg of protein.

Tissue sections on glass slides were then transferred onto metal plates placed in direct contact with dry ice to preserve sample integrity during UV irradiation. The tissue sections were exposed to UV light (λ = 254 nm) at a dose of 1 J/cm$^2$ in an XL 1500 UV Spectrolinker (Spectronics Corporation). After UV exposure, organ sections were recovered by scraping directly into 15 mL ice-cold lysis buffer (see composition below) supplemented with protease inhibitors (Roche, 11873580001) and RNase inhibitor (1:1000, produced at the Protein Expression and Purification Core Facility, EMBL Heidelberg). For the non-crosslinked control, every other section was lysed directly without exposure to UV; sections from four mice were pooled. The lysates were pipetted up and down several times to dislodge tissue sections. They were finally passed once through a 25-Gauge needle (BD Microlance, 300400) and seven times through a 27-Gauge needle (BD Microlance, 302200). The homogenates were snap frozen in liquid nitrogen and stored at −80 °C until use.

### eRIC: Capture of RNP complexes

See ref. 9 for a reference step-by-step protocol. Cell lysates were thawed at 37 °C, incubated for 15 min at 45 °C, cooled down on ice, and the debris was pelleted for 5 min at 16,000×g at 4 °C. The supernatant was transferred to 15 mL DNA LoBind tubes (Eppendorf, 0030122208) and complemented with 5 mM extra of dithiothreitol (DTT; Biomol, 04020.100). About 200 µL of each sample were taken as input, and the rest was mixed with 15 mg of capture probe-coupled beads previously equilibrated three times with three volumes of lysis buffer. The samples were incubated for 1 h at 37 °C with gentle rotation. The beads were then collected on a magnet, and the supernatant was transferred to a fresh 15 mL DNA LoBind tube for a second round of capture. After the capture, the beads were transferred to 5 mL DNA LoBind tubes (Eppendorf, 0030108310) and washed first with the lysis buffer, and then twice with each of the buffers 1, 2, and 3 (see composition below). Each wash was performed with 5 mL of the corresponding buffer for 5 min at 37 °C with gentle rotation. A "pre-elution" step was performed by incubating the washed beads with 220 µL of nuclease-free water (Ambion) for 10 min at 40 °C and 800 rpm. The bead suspension was then divided into two aliquots: 200 µL were used for RNase-mediated elution of RNA-bound proteins, and the rest (20 µL) was heat-eluted to recover the nucleic acid. The beads were collected on a magnet and the supernatant was discarded. For RNase elution of proteins, the beads were resuspended in 150 µL of 1× RNase buffer (see composition below) containing 5 mM DTT, 0.01% NP40, ~200 U RNase T1 (Sigma-Aldrich, R1003–100KU), and ~200 U RNase A (Sigma-Aldrich, R5503). Following a 60 min incubation at 37 °C, 800 rpm, the beads were collected on a magnet, and the eluate was transferred to a fresh tube. The eluate was placed once more on the magnet to eliminate any remaining beads, and safely stored on ice. Heat elution of RNA was performed by incubating the aliquoted beads in 15 µL of water for 5 min at 95 °C, 800 rpm. The beads were immediately collected on the magnet, and the supernatant recovered as quickly as possible to avoid a drop in temperature (with the consequent re-capture of RNP complexes). Any trace of beads was removed by a second round of collection, as explained. Corresponding eluates from the two consecutive rounds of capture were combined. Eluates obtained by RNase treatment (total volume: ~400 µL) were supplemented with 0.05% of SDS (2 µL of 10% SDS), concentrated at 45 °C to a volume of ~100 µL in a SpeedVac, snap frozen, and stored at −80 °C. Crosslinked eluates from two mice were combined for in-depth proteomic analyses.

Lysis buffer: 20 mM Tris-HCl (pH 7.5), 500 mM LiCl, 1 mM EDTA, 5 mM DTT, 0.5% (w/v) LiDS.

Buffer 1: 20 mM Tris-HCl (pH 7.5), 500 mM LiCl, 1 mM EDTA, 5 mM DTT, 0.1% (w/v) LiDS.

Buffer 2: 20 mM Tris-HCl (pH 7.5), 500 mM LiCl, 1 mM EDTA, 5 mM DTT, 0.02% (v/v) NP40.

Buffer 3: 20 mM Tris-HCl (pH 7.5), 200 mM LiCl, 1 mM EDTA, 5 mM DTT, 0.02% (v/v) NP40.

10× RNase buffer: 100 mM Tris-HCl (pH 7.5), 1.5 M NaCl

## Purification of non-poly(A) RNP complexes from eRIC supernatants

Poly(A) RNA-depleted supernatants of brain, kidney, and liver origin obtained after two consecutive rounds of eRIC were stored at −80 °C until use. Then non-poly(A) RNP complexes present in the supernatants were isolated using the 2 C silica-based solid-phase extraction method[44] with the following modifications. Two purification rounds with a total of 14 mL were performed with supernatants of brain and kidney origin. One purification with 2.5 mL of liver supernatant was performed. One non-crosslinked control and four crosslinked samples per organ were processed. Employed reagents belong to the Quick-RNA Miniprep Kit (Zymo Research, R1055) unless specified. Supernatants were combined with four volumes of RNA Lysis Buffer, mixed, deposited in a 20 mL syringe barrel, and passed through a Spin-Away Filter column applying pressure with a plunger. The flow-through was combined with 1 volume of 100% ethanol, mixed well, and loaded into a Zymo-Spin IIICG Column applying vacuum (Qiagen, QIAvac Vacuum System). Columns were centrifuged at 16,000×$g$ for 30 s at room temperature (rt). Flow-though was discarded. Columns were washed with 400 μL of RNA Wash Buffer, centrifuged at 13,000×$g$ for 30 s at rt, and then incubated for 15 min at rt with 80 μL of DNA digestion mix (75 μL of DNA Digestion Buffer, 5 μL of DNase I (1 U/μL)). Columns were washed with 400 μL of RNA Prep Buffer, centrifuged at 13,000×$g$ for 30 s at rt, washed with 700 μL of RNA Wash Buffer, centrifuged likewise, washed with 400 μL of RNA Wash Buffer, and centrifuged at 16,000×$g$ for 60 s at rt. To elute RNA and RNP complexes, columns were incubated with 100 μL (liver samples) or 75 μL (brain and kidney samples) of water for 5 min at rt and then centrifuged at 13,000×$g$ for 30 s at rt. Eluates from the two purification rounds (brain and kidney) were pooled. Eluates were brought to a volume of 146 μL with water, combined with 16.8 μL of 10x DNase buffer and 5 μL of TURBO DNase (Thermo Fisher, AM2238) and incubated 30 min at 37 °C. Samples were subjected to a second round of purification (Quick-RNA Miniprep Kit, Zymo Research, R1055). RNA and RNP complexes were eluted with 100 μL of water. About 100 μL of brain and kidney eluates, and 50 μL of liver eluate diluted in one volume of water, were combined with 10 μL of 10x RNase buffer (100 mM Tris-HCl (pH 7.5), 1.5 M NaCl, 0.5% (vol/vol) IGEPAL CA-630, 5 mM DTT, 0.2 μL RNase A and 0.5 μL RNase T1, and incubated at 37 °C for 1 h. The non-poly(A) RBP landscapes of the brain, kidney, and liver were analyzed using one non-crosslinked and two crosslinked samples (each of the latter resulting from the combination of two independent RNase-treated eluates) and 2 μg of captured RNA per sample. For in-depth characterization of the liver non-poly(A) RBPs, we used 40 μg of captured RNA per sample, one non-crosslinked and four independent crosslinked samples.

## RNA extraction, capillary electrophoresis, cDNA synthesis, and real-time quantitative PCR

The concentration of the captured RNA (heat-eluted) was estimated using a NanoDrop spectrophotometer (Thermo Fisher Scientific). About 10 ng of eluted RNA was analyzed using an Agilent 2100 Bioanalyzer System using the RNA 6000 Pico Kit, following the manufacturer's instructions. Total RNA in the input was extracted using Trizol LS (Thermo Fisher) and analyzed the same way.

For RT-qPCR analysis of RNA, ~100 ng of captured RNA or input RNA were treated with DNase I (Thermo Fisher), and reverse transcribed using SuperScript III (Life Technologies) together with random hexamers (Life Technologies), following the manufacturer's instructions. For qPCR analysis of DNA, the same reaction was performed in parallel but without the DNase and reverse transcriptase enzymes. Real-time qPCR was performed in a QuantStudio 6 Flex system (Life Technologies) using the SYBR Green PCR Master Mix (Life Technologies, 4309155) and the following primers (5′ to 3′, forward: f, reverse: r): *28S* rRNA (f: TTACCCTACTGATGATGTGTTGTTG, r: CCTGCGGTTCC TCTCGTA), *Actinb* (f: CGCGAGAAGATGACCCAGAT, r: TCACCGGAGTC

CATCACGAT), *Gapdh* (f: GTGGAGATTGTTGCCATCAACGA, r: CCCA TTCTCGGCCTTGACTGT) and *18S* rRNA (f: GAAACTGCGAATGGCTC ATTAAA, r: CACAGTTATCCAAGTGGGAGAGG).

## Western blot analysis

Proteins isolated from the liver by ex vivo eRIC after applying the indicated UV dose (Supplementary Fig. 1b) were separated by SDS-PAGE, transferred to a nitrocellulose membrane, and analyzed by western blot with the following antibodies at the indicated dilutions: ELAV-like protein 1 (ELAVL1)/Hu-antigen R (HuR) (Proteintech, 11910−1-AP, RRID:AB_11182183, 1:5000), Nucleolin (Ncl) (Abcam, ab50279, RRID:AB_881762, 1:1000), Beta-actin (Actb) (Sigma-Aldrich, A1978, RRID:AB_476692, 1:5000), and Histone H4 (Abcam, ab10158, RRI-D:AB_296888, 1:4000). As secondary antibodies anti-rabbit (Abcam, ab97051, RRID:AB_10679369, 1:5000) or anti-mouse (Abcam, ab6789, RRID: AB_955439, 1:5000) immunoglobulin G (IgG) horseradish peroxidase (HRP) were employed.

## Sample preparation for mass spectrometry (MS) and TMT labeling

eRIC and non-poly(A)RIC samples were concentrated to a volume of ~100 μL using a SpeedVac apparatus and treated with 10 mM DTT in HEPES buffer (50 mM HEPES, pH 8.5) for 30 min at 56 °C to reduce disulfide bridges in proteins. Reduced cysteines were then alkylated for 30 min at room temperature with 20 mM 2-chloroacetamide in HEPES buffer (protected from light). Samples were prepared using the SP3 protocol[53,54]. In short, equal volumes of two types of Sera-Mag Speed Beads (Thermo Scientific, Cat# 45152101010250 and Cat# 65152105050250) were combined, and 2 μL of the bead mix were added to each sample. Acetonitrile (HPLC/MS grade, TH Geyer) was then added to a final concentration of 50%, and samples were incubated off a magnet for 8 min. Beads were captured on a magnetic rack for 2 min, and the supernatant was removed. Beads were washed twice with 70% ethanol (analysis grade, Merck) and then once with 100% acetonitrile. Beads were reconstituted in digestion buffer (50 mM HEPES pH 8.5) supplemented with trypsin (sequencing grade, Promega) at an enzyme-to-protein ratio of 1:50 and incubated at 37 °C overnight. The digested peptides were recovered in HEPES buffer by applying two successive elution steps. The peptides were subsequently labeled with TMT10plex[55] Isobaric Label Reagent (Thermo Fisher) according to the manufacturer's instructions. The samples were combined and cleaned using an OASIS® HLB μElution Plate (Waters). Offline high pH reverse phase fractionation was carried out on an Agilent 1200 Infinity high-performance liquid chromatography system, equipped with a Gemini C18 column (3 μm, 110 Å, 100 × 1.0 mm, Phenomenex).

## Liquid chromatography with tandem mass spectrometry (LC −MS/MS)

An UltiMate 3000 RSLC nano-LC system (Dionex) fitted with a trapping cartridge (μ-Precolumn C18 PepMap 100, 5 μm, 300 μm i.d. × 5 mm, 100 Å) and an analytical column (nanoEase™ M/Z HSS T3 column 75 μm × 250 mm C18, 1.8 μm, 100 Å, Waters) was used. Trapping was carried out with a constant flow of trapping solution (0.05% trifluoroacetic acid in water) at 30 μL/min onto the trapping column for 6 min. Subsequently, peptides were eluted via the analytical column running solvent A (0.1% formic acid in water and 3% DMSO) with a constant flow of 0.3 μL/min, with an increasing percentage of solvent B (0.1% formic acid in acetonitrile and 3% DMSO) from 2 to 8% in 4 min, from 8 to 28% for a further 104 min, from 28 to 40% in another 4 min, and finally from 40 to 80% for 4 min, followed by re-equilibration back to 2% B in 4 min. The outlet of the analytical column was coupled directly to an Orbitrap Fusion Lumos Tribrid Mass Spectrometer (Thermo Scientific) using the Nanospray Flex ion source in positive ion mode.

The peptides were introduced into the Fusion Lumos using a Pico-Tip Emitter 360 μm OD × 20 μm ID; 10 μm tip (New Objective) and an applied spray voltage of 2.4 kV. The capillary temperature was set to 275 °C. A full mass scan was acquired with a mass range of 375–1500 m/z in profile mode in the orbitrap with a resolution of 120,000. The filling time was set to a maximum of 50 ms with a limitation of $4 \times 10^5$ ions. Data-dependent acquisition (DDA) was performed with the resolution of the Orbitrap set to 30,000, with a fill time of 94 ms and a limitation of $1 \times 10^5$ ions. A normalized collision energy of 38 was applied. MS2 data were acquired in profile mode.

## MS data analysis

IsobarQuant[56] and Mascot (v2.2.07) were used to process the acquired data, which was searched against the *Mus musculus* (UP000000589) Uniprot proteome database containing common contaminants and reversed sequences. The following modifications were included in the search parameters: Carbamidomethyl (C) and TMT10 (K) (fixed modification), acetyl (Protein N-term), oxidation (M), and TMT10 (N-term) (variable modifications). A mass error tolerance of 10 ppm and 0.02 Da was set, respectively, for the full scan (MS1) and MS/MS (MS2) spectra. Further parameters were: Trypsin digestion with a maximum of two missed cleavages tolerated; a minimum peptide length of seven amino acids, at least two unique peptides required for protein identification. The false discovery rate was set to 0.01 on both the peptide and protein level.

The protein output files of IsobarQuant were processed in R (ISBN 3-900051-07-0). Raw TMT reporter ion intensities ("signal_sum") were first cleaned for batch effects using the "removeBatchEffects" function of the limma package[57] and further normalized using the vsn (variance stabilization normalization) package[58]. Only proteins that were quantified with at least two unique peptides were considered for the analysis. Proteins were tested for differential expression using the limma package applying the normalization strategies indicated below. The replicate information was added as a factor in the design matrix given as an argument to the "lmFit" function of limma. A protein was annotated as a hit with a false discovery rate (FDR) smaller than 5% and a fold-change of at least 100% and as a candidate with an FDR below 20% and a fold-change of at least 50%.

**Ex vivo eRIC**. Two different normalization strategies were used, as indicated. In the first analysis, we estimated different normalization coefficients for each tissue and condition (+UV, −UV) of the eRIC samples. In the second approach (Supplementary Fig. 4), we estimated only a single normalization coefficient for +UV eRIC samples of the three tissues. Pearson's correlations (Fig. 1e and Supplementary Fig. 1c) were calculated using the cor function in R.

**Non-poly(A)RIC vs ex vivo eRIC of brain, kidney, and liver**. Proteins quantified in ex vivo eRIC or non-poly(A)RIC samples were considered. eRIC and non-poly(A)RIC samples were treated individually but evaluated in the same analysis. Different normalization coefficients were estimated for +UV and −UV conditions. To determine the relative association of RBPs with poly(A) and non-poly(A) RNA across organs (Fig. 4g), protein intensity was adjusted by RNA content per tissue (Fig. 3e).

**Comprehensive non-poly(A)RIC vs ex vivo eRIC of liver**. Two different normalization strategies were used, as indicated. In the first analysis, proteins quantified in ex vivo eRIC or non-poly(A)RIC in the liver were considered. eRIC and non-poly(A)RIC samples were treated individually but evaluated in the same analysis. Only the +UV samples were normalized in order to maintain the abundance difference from the -UV control. In the second approach (Supplementary Fig. 3d), only proteins quantified in both non-poly(A)RIC and eRIC in the liver were

considered. Non-poly(A)RIC and eRIC +UV samples (measured in different TMT experiments) were combined to look for relative differences between them.

## Global assessment of protein–protein crosslinking: in-gel digestion and sample preparation

Two consecutive rounds of ex vivo eRIC were performed as described above, using two kidneys per sample as starting material. Ex vivo eRIC eluates were vacuum-concentrated up to ~30 μL in a SpeedVac, combined with 10 μL of 4x Laemmli Sample Buffer (200 mM Tris-HCl pH 6.8, 40% (v/v) glycerol, 10% (v/v) beta-mercaptoethanol, 8% (w/v) SDS, 1.54% (w/v) DTT, 0.04% (w/v) bromophenol blue), incubated at 95 °C for 5 min and loaded into 4–15% acrylamide gels (Bio-Rad, 567-1083). Using as a reference the bands of a ladder (Bio-Rad, 1610374) loaded into the same gel, the lanes were cut using a scalpel at the following molecular weights (KDa): 20, 25, 37, 50, 75, 100, 150, and near the top of the gel. The isolated bands were subjected to in-gel digestion with trypsin. Peptides were extracted from the gel pieces by sonication for 15 min, followed by a quick spin and supernatant collection. A solution of 50:50 water: acetonitrile, 1% formic acid (2 x the volume of the gel pieces) was added for a second extraction, and the samples were again sonicated for 15 min, quickly spun down and the supernatant pooled with the first extract. The pooled supernatants were dried by vacuum centrifugation (SpeedVac). The samples were dissolved in 10 μL of reconstitution buffer (96:4 water: acetonitrile, 1% formic acid) and analyzed by LC-MS/MS.

## Global assessment of protein–protein crosslinking: LC-MS/MS

An UltiMate 3000 RSLC nano-LC system (Dionex) fitted with a trapping cartridge (μ-Precolumn C18 PepMap 100, 5 μm, 300 μm i.d. × 5 mm, 100 Å) and an analytical column (nanoEase™ M/Z HSS T3 column 75 μm × 250 mm C18, 1.8 μm, 100 Å, Waters) was used. Trapping was carried out with a constant flow of trapping solvent (0.05% trifluoroacetic acid in water) at 30 μL/min onto the trapping column for 6 min. Subsequently, peptides were eluted and separated on the analytical column using a gradient composed of Solvent A (0.1% formic acid in water and 3% DMSO) and solvent B (0.1% formic acid in acetonitrile and 3% DMSO) with a constant flow of 0.3 μL/min. The outlet of the analytical column was coupled directly to an Orbitrap Fusion Lumos Mass Spectrometer (Thermo Scientific) using the nanoFlex source.

The peptides were introduced into the Orbitrap Fusion Lumos via a Pico-Tip Emitter 360 μm OD × 20 μm ID; 10 μm tip (CoAnn Technologies) and an applied spray voltage of 2.4 kV, the instrument was operated in positive mode. The capillary temperature was set at 275 °C. Full mass scans were acquired for a mass range of 350–1500 m/z in profile mode in the orbitrap with a resolution of 120,000. The filling time was set to a maximum of 250 ms with a limitation of $4 \times 10^5$ ions. The instrument was operated in data-dependent acquisition (DDA) mode and MSMS scans were acquired in the Iontrap using a Rapid scan rate, with a fill time of up to 35 ms, and the AGC target was set to standard. A normalized collision energy of 30 was applied. MS2 data was acquired in centroid mode.

## Global assessment of protein–protein crosslinking: MS data analysis

The raw mass spectrometry data were processed with MaxQuant (v1.6.17.0)[59] and searched against the Uniprot *Mus musculus* database (UP000000589). The data were searched with the following modifications: Carbamidomethyl (C) (fixed modification), Acetyl (N-term), and Oxidation (M) (variable modifications). The mass error tolerance for the full scan MS spectra was set to 20 ppm and for the MS/MS spectra to 0.5 Da. A maximum of two missed cleavages was allowed. For protein identification, a minimum of one unique peptide with a peptide length of at least seven amino acids was required. A false

discovery rate below 0.01 was required on both the peptide and protein levels.

The raw output file of MaxQuant (proteinGroups.txt – file) was processed using the R programming language (ISBN 3-900051-07-0). Only proteins that were quantified with at least two unique peptides and in at least two out of three replicates in a specific gel fraction, were considered for the analysis. iBAQ values were cleaned for batch effects using the "removeBatchEffect" function of the limma package. In order to calculate the proportion of observation for each protein and sample (replicate and gel fraction), the corresponding iBAQ value was divided by the sum of all valid iBAQ values over all samples for the specific protein. The density distribution of observed molecular weights was weighted by this calculated proportion.

### poly(A) RNA and total RNA determination

To estimate the poly(A) RNA content, ex vivo eRIC heat eluates (see "eRIC: Capture of RNP complexes") from two consecutive rounds of capture were combined and the concentration of RNA quantified in a Nanodrop spectrophotometer (Thermo Fisher Scientific). The protein concentration of the eRIC lysates was determined using the DC Protein Assay (Bio-Rad) following the manufacturer's instructions. For the assessment of the total RNA content, a small piece of the organ of 20–45 mg was transferred to a 1.5 mL tube, and lysed and homogenized in 200 µL of RIPA lysis buffer (10 mM Tris-HCl pH 8, 150 mM NaCl, 1 mM EDTA, 1% NP40, and 0.1% SDS) using a plastic pestle. Half of the volume (100 µL) was employed to purify total RNA using the Quick-RNA Miniprep Kit (Zymo Research, R1055) following the manufacturer's instructions, and the isolated RNA was quantified in a Nanodrop spectrophotometer (Thermo Fisher Scientific). The other half of the lysate was employed to determine the protein concentration using the DC Protein Assay (Bio-Rad) following the manufacturer's instructions. Four independent measures per organ were performed.

### eRIC hit classification, gene ontology (GO), and domain analysis

Mouse and human RNA interactome studies, along with functional annotations, were downloaded from the RBPbase (https://rbpbase.shiny.embl.de, v.0.2.0). The R package "ggupset" was used for the visualization of overlaps between datasets with UpSet plots[60,61]. Fisher's exact with independent hypothesis weighting (IHW) for multiple hypothesis testing corrections was used for the overrepresentation analysis of protein domain information from MouseMine[62].

We compared RBPs detected in organs with total proteome data from cell lines used in previous RBP profiling studies. We extracted and mapped total proteome data for murine HL-1[31], MEF[63], mESC[64], RAW264.7[65] and human Jurkat[66], HEK293[66], HeLa[66], HuH7[67], U2OS[66] cell lines.

GO-term enrichment analysis in Fig. 2b were conducted with AmiGO 2[68] (http://amigo.geneontology.org/amigo), using the following parameters: analysis type: PANTHER overrepresentation test (Released 20200407); Annotation version and release date: GO ontology database Released 2020-02-21; reference list: *Mus musculus* (all genes in a database); test type: Fisher's exact with Bonferroni correction for multiple testing. Graphical representations were made with the ggplot2 R package[69]. KEGG- and remaining GO-term enrichment analysis were performed in g:Profiler (https://biit.cs.ut.ee/gprofiler/gost)[70,71], using the following parameters: organism: *Mus musculus*; statistical domain scope: only annotated genes; significance threshold: g:SCS threshold (tailor-made algorithm for multiple testing correction). In Fig. 6, the results were manually curated to exclusively select pathways of intermediary metabolism. Due to space constrains, a selection of terms was included in the figures. The corresponding full lists of GO-enriched terms can be found in Supplementary Data 8, 9.

PANTHER protein class analysis was performed using the PANTHER classification system[22] (http://www.pantherdb.org) (v.16.0). The percentages shown were calculated against the total number of protein class hits.

### Analysis of nucleotide-binding domains and cofactors

These analyses were performed on proteins listed in the PANTHER protein class: "metabolite interconversion enzyme" (enzyme). InterPro domain and cofactor annotations were retrieved from the UniProt mouse database (release-2021_03). Protein domains were classified as shown in Supplementary Data 10. Cofactors annotations were extracted from the sections "Catalytic activity", "Cofactor" and "Keywords". Enrichment of domains/cofactors was performed on enzymes that scored as eRIC hits in at least one organ, and compared to enzymes that were detected in inputs but not in eRIC eluates. The p-value was computed by Fisher's exact test, and corrected for multiple testing by the Benjamini–Hochberg method.

### Proximity ligation assay (PLA)

The PLA was adapted from a previously described protocol[23]. An anchor probe was designed to target the 3'UTR/poly(A) boundaries of mRNAs. The probe is composed (5' to 3') of a 5'biotin tag [BtnTg], a poly(dT) 18mer followed by two random nucleotides ("NN"). The control probe lacks the biotin tag but is otherwise identical.

Mouse organs (kidney, liver, and brain) were obtained as described above. 10-µm-thick sections were prepared in a cryostat (Leica Biosystems, Leica CM3050 S), transferred onto SuperFrost Plus adhesion slides (Carl Roth), encircled by a hydrophobic border with a 2-mm-thick pap pen (Sigma-Aldrich), and fixed with 4% paraformaldehyde at room temperature for 20 min. The fixed tissue sections were washed with PBS and permeabilized with PBS containing 1% BSA and 0.1% Triton X-100 at room temperature for 30 min. Tissue sections were then washed once for 5 min with 0.1 M Triethanolamine containing acetic anhydride and twice with PBS-T (0.02% Tween-20). The sections were further washed twice with hybridization buffer (1x Denhardt's solution, 0.1% (v/v) Tween-20, 0.1% (w/v) CHAPS, 5 mM EDTA, 1 mg/mL RNase free tRNA, 100 µg/mL heparin) and then incubated in hybridization buffer containing 100 nM of the probe in a wet chamber at 37 °C overnight. The probe was boiled for five minutes at 95 °C prior to addition. After hybridization, the sections were subjected to 5 min washes first in 50% (v/v) deionized formamide/5xSSC (saline-sodium citrate), then in 25% (v/v) deionized formamide/1xSSC, 12.5% (v/v) deionized formamide/2xSSC, 2xSSC/0.1% (v/v) Tween-20, and finally 0,2xSSC/0.1% (v/v) Tween-20. Subsequently, the Duolink PLA Fluorescence protocol (Sigma) was followed using an anti-biotin antibody (Abcam, ab201341, RRID:AB_2861249, mouse: 1:400) and either the anti-ENO1 (Proteintech, 11204-1-AP, RRID:AB_2099064, rabbit: 1:400), anti-SLC3A2 (Santa Cruz Biotechnology, sc-9160, RRID:AB_638288, rabbit: 1:400), anti-DDX6 (Novus Biologicals, NB200-192, RRID:AB_10000566, rabbit: 1:400) or anti-PKM1 antibody (Cell Signaling Technology, 7067, RRID:AB_2715534, rabbit: 1:400) for detection of the protein–RNA signal. The primary antibodies were incubated at room temperature for 90 min. After the last wash of the Duolink PLA Fluorescence protocol, the slides were incubated at room temperature for 45 min with antibody diluent mixed with DAPI (final concentration of 0.1 µg/µL) and nanobodies targeting rabbit IgG coupled with Alexa Fluor 488 (Chromotek, srbAF488-1-100, RRID:AB_2827585, alpaca nanobody). The tissue sections were washed once with PBS-T and a glass cover was mounted using ProLong Diamond Antifade Mountant (ThermoFisher Scientific, P36961). The prepared slides were stored at 4 °C until fluorescence microscopy.

Microscopy was performed using an LSM 780 Laser Scanning Microscope (ZEISS) equipped with an AxioCamera and a 63x/1.4 objective with immersion oil (Immersol 518 F, ZEISS, 10539438, Lot No. 170201). The microscope was operated using the ZEN 2012 software (ZEISS). The DAPI signal was recorded in one plane to act as a reference for counting the number of cells. The PLA (Alexa Fluor 594) and the

protein (Alexa Fluor 488 nanobody) signals were recorded as a Z-stack (ten pictures for a 10 μm stack). Two images were taken per tissue section from the brain, kidney, and liver of three different mice. The images were acquired as .lsm files using the same settings (gain, laser power, pinhole, and offset) for the same protein and analyzed using the Fiji software. The .lsm files were split into individual channels, the Z-stacks for the PLA signal and the Alexa Fluor 488 signal were projected into a single plane and the brightness was set to the same level in all images of the same channel to enable the comparison of the results. The individual channels were ultimately saved as .tiff files.

To count the PLA signals per cell for each of the different conditions, CellProfiler Software version 4.1.3 was used. The range of the signal spot size was set to 8 to 20 pixels and the range of the nuclear size (DAPI signal) was set to 100 to 400 pixels. In both instances, the global threshold strategy minimum cross-entropy was used. The threshold smoothing factor was 1.3488 for the PLA signals and 20 for the nuclei. The Alexa Fluor 488 signal was used as an outline of the cells. Clumped objects were separated by the intensity and objects touching the border of the images were discarded. The PLA signal per cell was counted by combining the information of the cellular outline (Alexa Fluor 488) and the DAPI signal. The statistical analysis of these results was performed using GraphPad Prism version 9 (one-way ANOVA with Tukey post hoc test).

### Reporting summary
Further information on research design is available in the Nature Portfolio Reporting Summary linked to this article.

## Data availability
The mass spectrometry proteomics data have been deposited to the ProteomeXchange Consortium via the PRIDE[72] partner repository with the dataset identifiers PXD032113 (ex vivo eRIC and total proteome of brain, kidney, and liver), PXD038076 (non-poly(A)RIC of brain, kidney, and liver), PXD038099 (comprehensive non-poly(A)RIC of the liver), and PXD038100 (global assessment of protein–protein crosslinks in ex vivo eRIC eluates of kidney origin). The RBP profiling studies used in this study are available in RBPbase (https://rbpbase.shiny.embl.de/) under the accession numbers RBPBASE000000007.1, RBPBASE000000008.1, RBPBASE000000009.1, RBPBASE000000010.1, RBPBASE000000012.1, RBPBASE000000013.1, RBPBASE000000032.1, RBPBASE000000033.1, RBPBASE000000034.1, RBPBASE000000035.1, RBPBASE000000036.1, RBPBASE000000037.1, RBPBASE000000038.1, RBPBASE000000039.1, RBPBASE000000040.1, RBPBASE000000041.1, RBPBASE000000046.1, RBPBASE000000047.1, RBPBASE000000048.1, RBPBASE000000049.1, RBPBASE000000050.1, RBPBASE000000051.1, RBPBASE000000059.1, RBPBASE000000060.1, RBPBASE000000061.1, RBPBASE000000062.1, RBPBASE000000066.1, RBPBASE000000067.1, RBPBASE000000014.1, RBPBASE000000016.1, RBPBASE000000017.1, RBPBASE000000018.1, RBPBASE000000019.1, RBPBASE000000053.1, RBPBASE000000054.1, RBPBASE000000055.1. More detailed access information for these studies, the RBPbase annotations, and the total proteome datasets used here are provided in Supplementary Data 11. Source data are provided with this paper.

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

## Acknowledgements
B.G. is supported by grants from the Deutsche Forschungsgemeinschaft (GA2075/3-1, GA2075/5-1, and GA2075/6.1). M.W.H. appreciates valuable support from MOLIT (Heilbronn, Germany) and the Manfred Lautenschläger Foundation. We thank Alfredo Castello (Glasgow), as well as Thileepan Sekaran from the Hentze laboratory, for his support with data analysis.

## Author contributions
J.I.P.-P., B.G., and M.W.H. conceived the project and designed the experiments. B.G. and D.F.-A. conducted the experimental work with help from J.I.P.-P. and I.H. J.I.P.-P. and T.S. performed data analysis with help from B.G. and S.S. M.R. and F.S. performed, respectively, the MS data acquisition and analyses. J.I.P.-P., B.G., and M.W.H. wrote the paper with input from all authors.

## Funding

## Competing interests
The authors declare no competing interests.
