## [Peer Review File · Nature Communications]

REVIEWER COMMENTS

Reviewer #1 (Remarks to the Author):

The manuscript by Perez-Perri and colleagues describes the RNA-binding protein landscape in mammalian organs and how this catalogue differs to the cultured cells. The work is well-written, concise, and results support the significance of specific findings although more functional follow-up would be make for a stronger manuscript.

The authors report the set of RBPs in three mice organs- liver, kidney and brain- and how this atlases compare to RBPs identified in cultured cells, providing the first RNA bound proteome of mammalian tissues. Comparison of the respective tissue-RBPomes allows identifying both overlapping RBPs, which are particularly prevalent in enzymes that bind nucleotide cofactors and tissue-specific RBPs. The results are very interesting and provide a significant amount of data that allow attractive speculation. In this study, the authors have established a novel protocol for crosslinking to investigate the spectrum of mRNA-binding proteins in tissues, overcoming an important limitation due to the poor penetration of UV light through the tissue. The conceptual advance provided by the manuscript, however is more modest. The presence of a specific set of RBPs in tissues is appealing, but the biological significance of this observation is not explored. The authors could provide further discovery, for example by testing the RNA targets of the 2 brain or kidney-specific RBPs identified in this study and the biological relevance of this interaction within the organ.

That said, there are still reasons to suspect that the identified mRNA binding proteins may be the result of experimental artefacts which somehow undermines the significance of the claims.

Specific comments:

Authors state that 'Irradiation of cells with ultraviolet (UV) light has been widely used because, unlike chemicals such as formaldehyde, UV exposure only exceptionally promotes protein-protein crosslinking and hence selects for direct RNA-protein interactions'. I disagree with this statement. It has been described that UV-crosslinking indeed promotes protein-protein interaction (please see Itri et al. *Cell Mol Life Sci* (2016)73(3):637-48. doi: 10.1007/s00018-015-2015-y; Leo et al. *Rapid Commun Mass Spectrom.* (2013). 27(14):1660-8. doi: 10.1002/rcm.6610. So, given that the slices are exposed to UV light ($\lambda=254$ nm) at a dose of 1 J/cm², it is not clear to which part effects are induced by high-crosslinking rates or indeed relate to organs and could explain why so many enzymes that may arrange in complexes come up as putative mRBPs. Authors indicate that they performed titration experiments to maximise RNA-RBP crosslinking efficiency while minimizing sample heating and preserving RNA integrity (although data is not shown) but experiments to rule out a possible protein-protein interactions should be performed.

Many of the concepts in this manuscript had been already posed by Castello et al. (*Cell*, 2012) and Baltz et al. (*Mol. Cell*, 2012). Even the link to co-factors was pointed out in Scherrer et al., 2010. *Plos ONE* (not cited). As it stands, the manuscript provides a valuable resource of RBPs that bind to mRNA in tissues. I agree with the authors in that the observation of a non-conserved, tissue-RNA-binding

proteome is new but, in my opinion, this is somehow expected given the plasticity of the RBPome upon different conditions even within the same organism. Functionality is, however, not evaluated, nor is the value of the RBP resource for novel biological discovery. I understand that exploration of the biological significance of hundreds of proteins is a daunting task, and I am not asking for this. I rather invite the authors to provide a proof of principle example supporting the biological value of their findings.

Reviewer #2 (Remarks to the Author):

General overview:

The provided manuscript by Hentze, Galy and colleagues describes a proteomics methodology that enables the UV crosslinking based interrogation of RNA binding proteins (RBPs) in mouse organs *ex vivo* at proteome scale (RBPome). Their workflow builds on their previously published eRIC technology, which utilizes LNA containing capture probes (targeting polyA RNAs) and is deemed to provide improved specificity and efficiency. In order to overcome the problem that penetration depth of UV light in mammalian organs is quite limited, the authors adopt and streamline an established technology from the field of electron microscopy namely cryosectioning of tissues. As a test case, they embarked on studying murine brain, liver and kidney, thereby significantly expanding the current list of mammalian RBP candidates identified by mass spectrometry in mammalian organs (towards this end there has been only one global approach published that employed formaldehyde (FA) crosslinking of mouse tissue: FAX-RIC, NAR 49, 2021). Technologically, this is an important step forward for the community. The proposed work is with no doubt a classical high quality piece of research (for which the Hentze laboratory is well known for) and is already written in a quite elegant way. However, after reading it my main conclusion is: "Yes, we know it!" For me the story does not trigger a single "heureka moment" and even worse, the oeuvre does not provide any real biological insight! Therefore, I would advise publishing the manuscript in a more specialized or methods focused journal (like *Molecular Systems Biology*, *Scientific Reports* or *npj systems biology and applications*). A more detailed argumentation is outlined in the following.

One of the major claims of Perez-Perri et al. is the discovery of enzymes from intermediary metabolism as pervasive RNA binders in the three mouse organs (under investigation). Until now, the dozens of UV- or FA-assisted global RBPome catalogues have identified a hundred of metabolic enzymes and the question remains if most of them are just bystanders (for instance RNA has a general role in preventing protein aggregation: *EMBO Reports* (2020)21:e49585) or really play a causative functional biological role. The finding of Perez-Perri et al. (figure 6) that especially enzymes harboring nucleotide binding domains represent a substantial fraction of their RNA interacting enzyme catalogue may point towards this direction. Of note, it has been proposed by Tony Hyman

that the cellular ATP concentration is critical for regulating protein phase separation, solubility and aggregation. Likewise, other nucleotide-containing cofactors or RNA might elicit similar effects. In addition, I do not even want to discuss the potential issues (identifying abundant metabolic enzymes) associated with substantial rRNA contamination in RBPome captures (also seen in this study: figure 1b and suppl. figure 1a)! The authors would also like to suggest that the literature already demonstrates/proves the widespread specific biological function of enzyme RNA interactions but the given citations (#35, #37, #38, #39, #44, #45, #46) reveal that most of this insight was gained by hypothesis-driven approaches, like in vitro assays (EMSA, FA-XL RNA-IP), ribosome-polysome analysis, RNA aptamer affinity pull-downs, or the much more specific and sophisticated RBDmap methodology (pioneered by the Hentze laboratory). Moreover, the fact that mammalian tissue RBPomes will be different as compared to standard cell culture RBPomes (HeLa, HEK293, mESCs etc.) is also not very surprising and similar observations were made in the context of multiprotein complexes (e.g. the groundbreaking work of A. Ori and M. Beck). Furthermore, because mammalian organ UV-XL RBPomics is technically challenging, the depth of analysis is severely hampered as compared to studies carried out in cellulo. Hence, the comparison of the organ RBPomes with the corresponding cellular data is of very limited value, because it faces a kind of “apple and oranges comparison problem” (figure 4b, figure 5a, supplementary figure 4a, b, c). However, I have to congratulate the authors, because it was very smart to choose brain as one of the organs since neurons are evolutionary well known for their widespread and diverse use of RNA-based regulatory processes (e.g. alternative splicing, RNA editing, extended 5’UTR and 3’UTR expansion in neuronal mRNAs; thus ample of opportunity to catch novel RBPs). Unfortunately, in brain glia cells are at least as abundant as neurons and many of these neuronal RNA processes take place in a quite localized manner (axons etc.). Therefore, it will be very difficult to capture them by the current sensitivity of the presented technology (only two brain-specific candidate RBPs are discovered, figure 3a; but at least the GO cellular component AU-rich element binding is enriched, figure 2b).

So as to gain more functional biological insight the Hentze team could attempt several experiments. Scale up the study, thereby increasing the depth of analysis, which would allow them to conduct RBDmap. The latter will uncover the RNA binding domains or regions of the RNA interacting metabolic enzymes and presumably disclose some common or general binding/regulatory mechanism. Alternatively, an iCLIP or similar type of protein-centric methodology selecting a couple of RBP candidate metabolic enzymes might give a deeper biological insight based on the type and nature of bound RNAs. Another option would be to team up with metabolomics experts and scrutinize a possible correlation between the different organ metabolomes (using a cell culture metabolome as control) and the type of RNA interacting metabolic enzymes.

Major points:

-It is not really clear to me how many (technical) replicates were measured by nLC-MS? The authors admit that only two biological replicates of UV-irradiated mice were employed (each of them representing a pool of two individual mice) in conjunction with a single pool of four non-UV control

mice. I guess that many cryosection specimens can be created out of one organ? Hence, a more detailed description of the data analyses and statistical validation workflows is required.

-Figure 1a, right panel and methods: how is the total protein normalization achieved in more detail? By measuring an aliquot of each sample before mixing or by mixing aliquots of the samples with different mixing ratios?

-Is it possible to determine the total protein and RNA content of individual organ cryosection specimens?

-Figure 1b and supplementary figure 1a: it is highly advisable to characterize the RNAs by deep RNAseq (possibly with and without “ribodepletion”), which became an easy, straightforward and cheap QC technology that will even reveal the content of polyA harboring lncRNAs.

Minor points:

-Discuss mRNA stability/half live in brain, kidney and liver.

-Figure 1e: please add the same scatter plots exclusively considering RBP harboring proteins

-Figure 3b: it looks like that UV crosslinking in brain works less efficient as compared to the other tissues; please discuss why.

-Figure 3d: maybe label the figure with proximity labeling assay.

-Figure 3c and supplementary figure 3b: the log₂ ratio is most likely referring to the ratio plus/minus UV?

-Supplementary figure 1b and 3c: please catalogue the eRIC specific proteins (with the artificially imputed value) in a separate table.

-Please describe the “Hentze lab” SP3 workflow in more detail, e.g. which organic solvent was employed for bead protein aggregation (acetonitrile, ethanol, propanol)?

-Instead of harnessing the total proteome, using the list of expressed genes/proteins (in a given organ; should be available from RNAseq data) is usually a much better “background proteome” for GO analyses.

Reviewer #3 (Remarks to the Author):

This manuscript from Perez-Perri, et al. describes a new method (“eRIC”) to identify proteins binding to poly(A)+ RNA in mouse organs. For a variety of technical reasons, including the inability for UV to penetrate tissues, this aspect of RNA biology has been unexplored, especially relative to our knowledge of proteins binding RNA in cultured cells. One strength of this manuscript is the development of this method (which involves tissue slicing and robust technical controls), which will also have applications to other UV-based methods. Another strength of this method are the associated datasets, which will be of interest to the RNA community. For the most part, this manuscript is clearly presented, and the interpretations are not overstated. The weakest part of the manuscript are the analyses on the differences between RBPs bound in the three different organs and in tissue culture cells. There are additional alternative interpretations, and the data (and discussion) can be a little hard to follow at times. Given that the following issues are addressed, I support publication of this study:

1. For the new RBPs identified (i.e., those that have not shown up in previous RIC studies), how many are expressed in tissue culture cells? The types of analyses performed in Figure S4 would be useful for this class of RBPs as well.
2. The observation of altered binding of RBPs is very intriguing. However, this section would be stronger if the authors could also exclude some additional alternative explanations, such as reduced poly(A)+ RNA concentration in cells or altered protein localization.
3. I was a little confused by the statement that many (how many?) RBPs did have different expression. A clearer analysis of the data would really help the reader.

Minor comments:

1. P1, line 49: “only exceptionally promotes protein-protein crosslinking” is unclear.
2. Some of the word usage is awkward, and some additional editing would help.

REVIEWER COMMENTS

Reviewer #1 (Remarks to the Author):

The manuscript by Perez-Perri and colleagues describes the RNA-binding protein landscape in mammalian organs and how this catalogue differs to the cultured cells. The work is well-written, concise, and results support the significance of specific findings although more functional follow-up would be make for a stronger manuscript.

The authors report the set of RBPs in three mice organs- liver, kidney and brain- and how this atlases compare to RBPs identified in cultured cells, providing the first RNA bound proteome of mammalian tissues. Comparison of the respective tissue-RBPomes allows identifying both overlapping RBPs, which are particularly prevalent in enzymes that bind nucleotide cofactors and tissue-specific RBPs. The results are very interesting and provide a significant amount of data that allow attractive speculation. In this study, the authors have established a novel protocol for crosslinking to investigate the spectrum of mRNA-binding proteins in tissues, overcoming an important limitation due to the poor penetration of UV light through the tissue. The conceptual advance provided by the manuscript, however is more modest. The presence of a specific set of RBPs in tissues is appealing, but the biological significance of this observation is not explored. The authors could provide further discovery, for example by testing the RNA targets of the 2 brain or kidney-specific RBPs identified in this study and the biological relevance of this interaction within the organ.

We thank this referee for the positive evaluation of the quality and significance of our work. This manuscript mostly addresses two aims: 1) to establish a technique that empowers researchers to determine the RNA-bound proteomes of intact organs under different conditions (including human clinical samples); 2) provision of the first comprehensive RBP atlases from mammalian organs, along with an in-depth analysis of the corresponding datasets with those of cultured cells. The latter results in a number of intriguing, unexpected (and validated) observations that open future directions of research. We agree that the paper lacks strictly functional data. These would be beyond the scope of this work, because the identification of high-confidence RNA targets of specific RBPs *ex vivo* and elucidating the biological relevance of these interactions represents a non-trivial and time-demanding task. This article rather lays the groundwork for detailed investigations of the biology of specific RBPs in organs.

The global and tissue-specific regulation of RNA-protein interactions represents one of the intriguing phenomena that we report (Figure 3). In revision, we have added additional depth to our analyses, determining the non-poly(A) RNA-bound proteomes of kidney, brain and liver (data presented in the new Figure 4; Supplementary Fig. 3, Supplementary Data 3 and Supplementary Data 4, also see below).

That said, there are still reasons to suspect that the identified mRNA binding proteins may be the result of experimental artefacts which somehow undermines the significance of the claims.

Reviewer 1 unfortunately does not specify the reasons for this statement. If this is meant as a general reminder that any system-wide approach is likely to yield at least a small number of false positives, we would agree. However, it is remarkable that 10 years after the first RNA interactome capture (RIC) studies were published, the data and conclusions have overwhelmingly been confirmed. Nonetheless, to directly address the possible risk of protein-protein crosslinking which could contribute “piggy-back riders” as false positives to our datasets, we evaluated the formation of protein-protein crosslinks in our experiments in a comprehensive and unbiased manner (see new Supplementary Fig. 5, Supplementary Fig. 6 and Supplementary Data 6). Our new data confirm that *ex vivo* RIC detects genuine RBPs (see the detailed answer below).

A further potential concern is that highly abundant proteins could nonspecifically co-isolate with poly(A) RNA, although we previously demonstrated that the utilization of LNA-modified oligo(dT) probes minimizes such contaminants (Perez-Perri et al., Nat. Comms, 2018). In revision, we have extended the eRIC approach to *ex vivo* organ samples to identify RBPs associated with non-poly(A) RNA (new Figure 4 and Supplementary Fig. 3). For this, RNP complexes were extracted from eRIC supernatants with guanidinium thiocyanate-phenol-chloroform and purified over a silica matrix (Asencio et al., Life Sci Alliance, 2018). As eRIC supernatants are depleted of poly(A) RNA by oligo(dT) bead selection, they contain essentially non-poly(A) RNP complexes. We identified 404 non-poly(A) RNA binders that were not present in eRIC eluates. Among the non-poly(A) RNA binders, 51 were ribosomal proteins. Ribosomal proteins are some of the most abundant RBPs in the cell. The finding that 51 ribosomal proteins are detected in the non-poly(A) RNP fraction but absent in the eRIC eluates shows that the poly(A) RBPs identified by *ex vivo* eRIC do not simply correspond to very abundant proteins (such as ribosomal proteins) and provides further experimental evidence of the high specificity of our method.

It is also important to note that the original RIC protocol, from which *ex vivo* eRIC is derived, led to the discovery of hundreds of non-canonical RBPs. These non-canonical RBPs lack documented RNA-binding domains or known RNA-related functions and have therefore been suspected to represent "false positives". However, over the past years these unconventional RBPs have been identified by different laboratories around the world using independent molecular techniques and in different organisms. A growing number of non-canonical RBPs have meanwhile been assigned clear biological functions directly related to their ability to bind RNA. This includes, amongst others, ACO1, GAPDH, FASTKD2, p62, HSD17B10, PKM2, and ENO1. The data added during this revision and recent reports in the literature further support the notion that the reported *ex vivo* eRIC datasets from mouse organs largely represent genuine RBPs whose RNA-related biological functions are yet to be discovered. The method and resources described in this manuscript will contribute towards this goal.

Specific comments:

Authors state that 'Irradiation of cells with ultraviolet (UV) light has been widely used because, unlike chemicals such as formaldehyde, UV exposure only exceptionally promotes protein-protein crosslinking and hence selects for direct RNA-protein interactions'. I disagree with this statement. It has been described that UV-crosslinking indeed promotes protein-protein interaction (please see Itri et al. Cell Mol Life Sci (2016)73(3):637-48. doi: 10.1007/s00018-015-2015-y; Leo et al. Rapid Commun Mass Spectrom. (2013). 27(14):1660-8. doi: 10.1002/rcm.6610. So, given that the slices are exposed to UV light ($\lambda=254$ nm) at a dose of 1 J/cm², it is not clear to which part effects are induced by high-crosslinking rates or indeed relate to organs and could explain why so many enzymes that may arrange in complexes come up as putative mRBPs. Authors indicate that they performed titration experiments to maximise RNA-RBP crosslinking efficiency while minimizing sample heating and preserving RNA integrity (although data is not shown) but experiments to rule out a possible protein-protein interactions should be performed.

We agree that Leo et al. demonstrated that aromatic amino acid side chains can promote the formation of covalent bonds with nearby residues upon exposure to ultrashort UV laser pulses. Moreover, Itri *et al.*, could detect cross-linked products of GAPDH in live HeLa cells subjected to UV irradiation. Hence, although previous RNA-interactome studies in cultured cells suggest that protein-protein UV crosslinking is uncommon (Castello et al., 2012), the studies of Leo and Itri indicate that such crosslinking reactions can occur, calling for caution when interpreting RIC data. To address the concern experimentally, the proteins captured by *ex vivo* eRIC were separated according to their molecular weight (MW) by polyacrylamide gel electrophoresis. The gel was divided into 7 fractions, each corresponding to a defined MW range, and the fractions were analyzed by mass spectrometry (MS). We then determined the distribution of each protein across the different fractions. Since RNA is degraded during elution as part of

the eRIC protocol, proteins should shift to higher MW fractions when they have formed protein-protein crosslink complexes. We applied this test to eRIC eluates from kidney. The data are presented in the new Supplementary Fig. 5, Supplementary Fig. 6 and Supplementary Data 6 of the revised manuscript. They reveal that the vast majority of proteins identified by MS are present in fractions corresponding to their predicted monomeric molecular mass and thus represent *bona fide* RBPs. Proteins that were detected in higher MW fractions, especially the top two, either run close to the cutting position or are actually not eRIC hits (Supplementary Fig. 6 and Supplementary Data 6). Only ~7.5% of the detected proteins that localize to higher MW fractions are eRIC hits and have a native molecular mass of at least 10 KDa less than the cutting position. Of note, even this small percentage likely overestimates “false positives”, since many proteins that shifted to higher molecular mass fractions are actually validated RBPs (e.g Rbm14, Fus, Ncl, Hnrnpul2, etc). Overall, the new data further support that UV crosslinking between proteins hardly contributes to candidate RBP assignments under our experimental conditions. To the best of our knowledge, such global assessment of protein-protein UV crosslinking has not been reported before, and therefore adds value to this revision. The reviewer also refers to the UV conditions used in *ex vivo* eRIC. We now present the data showing a titration of UV dose. As shown in new Supplementary Figure 1c, a dose of 1 J/cm² allowed maximum recovery of two well-known RBPs (HuR and Ncl) in eRIC eluates, while negative controls (beta actin and histone H4) were not detected.

Many of the concepts in this manuscript had been already posed by Castello et al. (Cell, 2012) and Baltz et al. (Mol. Cell, 2012). Even the link to co-factors was pointed out in Scherrer et al., 2010. Plos ONE (not cited). As it stands, the manuscript provides a valuable resource of RBPs that bind to mRNA in tissues. I agree with the authors in that the observation of a non-conserved, tissue-RNA-binding proteome is new but, in my opinion, this is somehow expected given the plasticity of the RBPome upon different conditions even within the same organism. Functionality is, however, not evaluated, nor is the value of the RBP resource for novel biological discovery. I understand that exploration of the biological significance of hundreds of proteins is a daunting task, and I am not asking for this. I rather invite the authors to provide a proof of principle example supporting the biological value of their findings.

As explained above, the revised manuscript extends the state-of-the art in RBP biology in several ways: 1) It establishes an enabling method to study RBPs in mammalian organs, in principle applicable to clinical specimen; 2) it provides the first comprehensive atlases of RBPs of three mouse organs (including the first RBPome of neurological origin overall, i.e. also including cell-based studies); 3) it adds further experimental evidence that non-canonical RBPs (including enzyme-RBPs) identified in RIC studies are genuine RNA binders; 4) it reveals differences in RBP activity across organs and between organs and cultured cells; 5) it implicates global regulation of RBP activity across organs, which to the best of our knowledge was neither known nor evident.

Large scale organ-specific regulation of RBP activity (kidney > liver > brain, Figure 3) is potentially of great biological significance. We previously showed that global changes in RBP activity do not correlate with RBP abundance, and do not reflect differences in the capture, integrity, or purity of poly(A) RNA/RBP complexes, or reflect variations in UV cross-linking efficiency or sample processing and MS analysis. We suggested that the overall changes in RBP activity could be explained by differences in poly(A) RNA content relative to the amount of protein present in the tissue. Consistent with this idea, we now report that poly(A) RNA levels are lower in the brain than in kidney and liver (see new Figure 3e in the revised manuscript). However, we observe an inverse correlation between RBP activity and the amount of poly(A) RNA in the kidney compared to liver. RBP activity was higher in kidney than in liver, while poly(A) RNA levels were lower. How to explain such differences? CLIP and CLIP-like experiments revealed that RBPs, such as e.g. SERBP1 (Backlund et al., Nucl. Acids Res., 2020), can differentially interact with poly(A) RNA or non-poly(A) RNA in response to biological cues. On the other side, we found that non-poly(A) RNA levels are higher in liver

than in kidney (Figure 3e). We refined the eRIC methodology to isolate and characterize proteins bound to non-poly(A) RNA in brain, kidney and liver, and compared the data to our previous poly(A) RBP datasets. We identified 222 proteins that associate with both poly(A) and non-poly(A) RNA (so-called dual binders, new Figure 4). We then compared the relative amount of these dual binders in poly(A) and non-poly(A) RNA isolates across tissues. Compared with the kidney, dual binders in the liver appear to interact more with non-poly(A) RNA and less with poly(A) RNA. (new Figure 4g, left two panels). This is not the case for 583 proteins that bind exclusively to poly(A) RNA (Figure 4g, right panel), or for 133 selective non-poly(A) RNA binders (Figure 4g, third panel from the left). These additional data add further value to the revised manuscript.

Reviewer #2 (Remarks to the Author):

General overview

The provided manuscript by Hentze, Galy and colleagues describes a proteomics methodology that enables the UV crosslinking based interrogation of RNA binding proteins (RBPs) in mouse organs *ex vivo* at proteome scale (RBPome). Their workflow builds on their previously published eRIC technology, which utilizes LNA containing capture probes (targeting polyA RNAs) and is deemed to provide improved specificity and efficiency. In order to overcome the problem that penetration depth of UV light in mammalian organs is quite limited, the authors adopt and streamline an established technology from the field of electron microscopy namely cryosectioning of tissues. As a test case, they embarked on studying murine brain, liver and kidney, thereby significantly expanding the current list of mammalian RBP candidates identified by mass spectrometry in mammalian organs (towards this end there has been only one global approach published that employed formaldehyde (FA) crosslinking of mouse tissue: FAX-RIC, NAR 49, 2021). Technologically, this is an important step forward for the community. The proposed work is with no doubt a classical high quality piece of research (for which the Hentze laboratory is well known for) and is already written in a quite elegant way. However, after reading it my main conclusion is: "Yes, we know it!" For me the story does not trigger a single "heureka moment" and even worse, the oeuvre does not provide any real biological insight! Therefore, I would advise publishing the manuscript in a more specialized or methods focused journal (like Molecular Systems Biology, Scientific Reports or npj systems biology and applications). A more detailed argumentation is outlined in the following.

We thank the reviewer for the positive comments on the quality of our work. As detailed above in our responses to reviewer #1, we believe that this work makes substantial contributions that advance the state-of-the-art in ways that qualify it for a broad interest journal. Mechanistic functional data are lacking, because they would "explode" the timeframe and scope of a single publication.

One of the major claims of Perez-Perri et al. is the discovery of enzymes from intermediary metabolism as pervasive RNA binders in the three mouse organs (under investigation). Until now, the dozens of UV- or FA-assisted global RBPome catalogues have identified a hundred of metabolic enzymes and the question remains if most of them are just bystanders (for instance RNA has a general role in preventing protein aggregation: EMBO Reports (2020)21:e49585) or really play a causative functional biological role. The finding of Perez-Perri et al. (figure 6) that especially enzymes harboring nucleotide binding domains represent a substantial fraction of their RNA interacting enzyme catalogue may point towards this direction. Of note, it has been proposed by Tony Hyman that the cellular ATP concentration is critical for regulating protein phase separation, solubility and aggregation. Likewise, other nucleotide-containing cofactors or RNA might elicit similar effects.

We agree with the reviewer that RNA-protein interactions can play different roles and that more are emerging. For example, we recently reported that the enzymatic activity of enolase 1 is directly controlled by RNA and that this riboregulation is critically relevant for mouse embryonic stem cell differentiation (Huppertz et al., Mol. Cell, 2022). The candidate RBPs identified by *ex vivo* eRIC could therefore have different RNA-binding functions. Modulation of protein aggregation could well be one of them, and we previously reported that nearly half of the RNA-binding regions of RBPs fall into disordered domains (Castello et al., Mol Cell, 2016).

In addition, I do not even want to discuss the potential issues (identifying abundant metabolic enzymes) associated with substantial rRNA contamination in RBPome captures (also seen in this study: figure 1b and suppl. figure 1a)!

We previously reported that rRNA accounts for about 3% of the RNA captured by eRIC as opposed to RIC (PMID: 30352994). We do not consider this substantial, especially since rRNA represents ~ 90% of cellular transcripts. Previous attempts to completely remove rRNA from eRIC eluates have been associated with concomitant and proportional loss of poly(A) RNA (PMID: 30352994). The reasons for the presence of rRNA in eRIC samples remain unclear. The rRNA could be co-isolated due to RNA-RNA interactions with poly(A) transcripts. The addition of poly(A)-rich tails to the 3' end of rRNA, as observed in yeast and human cells (PMID: 15173578, PMID: 16738135, PMID: 20368444), could also mediate specific (!) binding to the oligo(dT) probe. Data added in revision and discussed above (responses to reviewer #1) further address the issue raised by the reviewer.

Regarding the role of protein abundance on eRIC data, we show that hundreds of proteins that are very abundant in the inputs are not detected in eRIC eluates (Supplementary Fig. 1b). In addition, we have now characterized the non-poly(A) RNA interactome of organs (see Figure 4, Supplementary Fig. 3 and section "Organ-specific interactions of RBPs with poly(A) and non-poly(A) RNA" on pages 4 and 5 of the revised manuscript), and found that many ribosomal proteins (51), which represent some of the most abundant RBPs, are expectedly identified as non-poly(A) binders, but are not detected in eRIC eluates. While we do not exclude that a few particularly abundant proteins may be false positives, our results overall indicate that *ex vivo* eRIC captures RBPs and more particularly poly(A) RBPs with high specificity.

The authors would also like to suggest that the literature already demonstrates/proves the widespread specific biological function of enzyme RNA interactions but the given citations (#35, #37, #38, #39, #44, #45, #46) reveal that most of this insight was gained by hypothesis-driven approaches, like *in vitro* assays (EMSA, FA-XL RNA-IP), ribosome-polysome analysis, RNA aptamer affinity pull-downs, or the much more specific and sophisticated RBDmap methodology (pioneered by the Hentze laboratory).

As mentioned by the referee, the biological roles of several enzyme-RNA interactions have been uncovered using entry points different from RIC. Unlike more historical methods, RIC as has only been available for about 10 years. The critical point is that RIC and especially eRIC experiments (re-)identify enzyme RBPs for which biological roles are clearly established. This also includes enzymes such as ENO1, SAHH, PKM2, or FASTKD2, for which the biological role of RNA binding has only recently been determined.

Moreover, the fact that mammalian tissue RBPomes will be different compared to standard cell culture RBPomes (HeLa, HEK293, mESCs etc.) is also not very surprising and similar observations were made in the context of multiprotein complexes (e.g. the groundbreaking work of A. Ori and M. Beck).

Our starting hypothesis was that organ and cell culture RBPomes would largely overlap, with possibly some systematic differences that would be rather difficult to guess. Our experiments therefore did not aim to uncover surprises, but to experimentally determine the facts. *Ex vivo* eRIC allowed for the first time the system-wide comparison of RBPomes from organs and cultured cells. We agree with the reviewer that differences between the RBPome of tissues and the RBPome of cultured cells are to be expected. Although technical aspects also have to be considered, we think that at least some of the differences between the RBPomes of organs and cultured cells represent interesting biological observations. They may, for example, reflect the influence of the metabolic environment or oxygen tension, or be attributable to the often cancerous origin and high proliferation rate of cultured cells, whereas intact tissues are non-transformed and essentially quiescent. We thus provide experimental datasets to explore the role of these parameters on RNA-protein interactions in the future. In the revised version of the manuscript we have assessed whether the novel RBPs identified in organs (Figure 5a-c) are at all expressed in the cell lines used in previous RBP profiling studies. Remarkably, we have found that 263 out of the 291 (>90%) novel RBPs are expressed in at least one of ten tested cell lines, with 42 RBPs (>14%) expressed in all of them (Figure 5d).

Furthermore, because mammalian organ UV-XL RBPomics is technically challenging, the depth of analysis is severely hampered as compared to studies carried out *in cellulo*. Hence, the comparison of the organ RBPomes with the corresponding cellular data is of very limited value, because it faces a kind of “apple and oranges comparison problem” (figure 4b, figure 5a, supplementary figure 4a, b, c).

UV crosslinking of RNA-protein interactions in intact tissues is indeed non-trivial, and no in-depth RIC data from mammalian organs were reported since the *in cellulo* work by Castello et al., and Baltz et al. about a decade ago. *Ex vivo* eRIC alleviates the bottleneck of low UV penetration into tissues and has allowed the identification of about 1300 RBPs. This number is perfectly in line with the yield of typical eRIC studies in cultured cells. *Ex vivo* eRIC not only detects most of the canonical RBPs that make up the core of the mammalian cell RBPome, but also identifies previously unknown RBPs. We think that this shows that the depth of mammalian organ RBPomics is fully in line with the earlier cultured cell studies, and hence argues against them being “severely hampered”. As discussed above, we also think that comparisons between organ and cell-based RBPomes have been perfectly possible and have yielded informative first results.

However, I have to congratulate the authors, because it was very smart to choose brain as one of the organs since neurons are evolutionary well known for their widespread and diverse use of RNA-based regulatory processes (e.g. alternative splicing, RNA editing, extended 5'UTR and 3'UTR expansion in neuronal mRNAs; thus ample of opportunity to catch novel RBPs). Unfortunately, in brain glia cells are at least as abundant as neurons and many of these neuronal RNA processes take place in a quite localized manner (axons etc.). Therefore, it will be very difficult to capture them by the current sensitivity of the presented technology (only two brain-specific candidate RBPs are discovered, figure 3a; but at least the GO cellular component AU-rich element binding is enriched, figure 2b).

We agree with these considerations. Our *ex vivo* eRIC study performed with whole tissue samples is a first step towards the exploration of the RBPome of intact organs, including the brain. Further refinement of the method may help studying the RBPome of specific tissue substructures in the future, especially also with the ongoing refinements in sensitive protein identification and quantification by mass spectrometry.

So as to gain more functional biological insight the Hentze team could attempt several experiments. Scale up the study, thereby increasing the depth of analysis, which would allow them to conduct RBDmap. The latter will uncover the RNA binding domains or regions of the RNA interacting metabolic enzymes and presumably disclose some common or general binding/regulatory mechanism. Alternatively, an iCLIP or similar type of protein-centric methodology selecting a couple of RBP candidate metabolic enzymes might give a deeper biological insight based on the type and nature of bound RNAs. Another option would be to team up with metabolomics experts and scrutinize a possible correlation between the different organ metabolomes (using a cell culture metabolome as control) and the type of RNA interacting metabolic enzymes.

We thank the reviewer for these relevant suggestions. Conducting RBDmap in tissues, defining the RNA interactome of specific enzyme-RBPs using CLIP, or looking for potential correlations between the metabolome and RBPome of different organs could and likely will provide additional information (as well as complexity). However, we think that these extensions are beyond the scope of the current study. Therefore, we have focused our additional experimental work on the validation of the method and reported datasets rather than on broadening the scope of the work. Nevertheless, we agree that these studies represent interesting avenues for future research.

Major points:

-It is not really clear to me how many (technical) replicates were measured by nLC-MS? The authors admit that only two biological replicates of UV-irradiated mice were employed (each of them representing a pool of two individual mice) in conjunction with a single pool of four non-UV control mice. I guess that many cryosection specimens can be created out of one organ? Hence, a more detailed description of the data analyses and statistical validation workflows is required.

As requested by the referee, a more detailed description of the protocol and data analysis has been included in results and the materials and methods sections of the revised manuscript (see sections "Refinement of eRIC to characterize the poly(A) RNA-bound proteomes of mammalian organs", and "MS data analysis" in pages 2 and 16, respectively). Regarding the amount of material needed for *ex vivo* eRIC, 30 μm cryosections are placed on a glass slide and cover an area of approximately 7 cm^2 (~60% of the area of a 4.6x2.5 cm glass slide). For the brain, this represents about 15-20 tissue sections (in the horizontal plane) per slide. For liver, this corresponds to 10-15 slices; for kidney, 20-30 sections. We determined that deep RBPomes can be obtained using about 30 glass slides per sample, corresponding to a surface of about 200 cm^2 of 30 μm thick tissue sections (equivalent to 1.5 brains, 2/3rd of the left lateral lobe of a liver, and 3 kidneys, respectively).

-Figure 1a, right panel and methods: how is the total protein normalization achieved in more detail? By measuring an aliquot of each sample before mixing or by mixing aliquots of the samples with different mixing ratios?

To compare RBPomes from different organs, normalization prior to MS analysis was performed on the basis of the amount of RNA captured, as in previous RIC studies. We used the amount of captured RNA for normalization, because it can be precisely measured (the mass of RNA that was submitted for each sample is shown in Figure 3b). By contrast, the determination of protein content is less accurate, because the absolute mass of captured protein is very low and typically below the detection limit of standard protein quantification methods (e.g. Bradford, BCA, etc.). Performing a pre-MS analysis of a small aliquot (<10%)

of each sample to estimate the protein content, and combining samples in a mixing ratio to obtain the same amount of protein per sample would represent an alternative.

-Is it possible to determine the total protein and RNA content of individual organ cryosection specimens?

The following numbers were extrapolated from the amount of cryosections used and the quantity of RNA and protein present in pooled samples:

Brain: 41.6 ng of poly(A) RNA per section, 342 µg of protein per section.

Kidney: 35.5 ng of poly(A) RNA per section, 94.4 µg of protein per section.

Liver: 113.4 ng of poly(A) RNA per section, 304.2 µg of protein per section.

We have not included this information in our revised manuscript, but we would be happy to do so, if requested.

-Figure 1b and supplementary figure 1a: it is highly advisable to characterize the RNAs by deep RNAseq (possibly with and without “ribodepletion”), which became an easy, straightforward and cheap QC technology that will even reveal the content of polyA harboring lncRNAs.

We estimated the levels of RNA captured qualitatively by capillary electrophoresis (Supplementary Fig. 1a) and quantitatively by qPCR (Fig. 1b). While we agree that deep RNAseq could be performed on the samples, we do not think that this would add to the data essential information, although one may well consider doing so in future studies. Since *ex vivo* eRIC isolates polyadenylated RNAs in an unbiased way, it is fair to expect that polyadenylated lncRNAs will contribute to the overall RBPomes.

Minor points:

-Discuss mRNA stability/half live in brain, kidney and liver.

To our knowledge, no system-wide studies of RNA turnover rates in mouse brain, kidney, and liver have been performed. We report RBPs bound to the steady state transcriptomes within the respective tissues. These include RBPs previously implicated in the control of RNA stability/half lives.

-Figure 1e: please add the same scatter plots exclusively considering RBP harboring proteins. Figure 1e shows the normalized signal sum of proteins present in eRIC eluates and thus crosslinked to RNA. The scatter plots already present RBP data only. Or did the reviewer mean RNA-binding domain (RBD)-harboring proteins?

-Figure 3b: it looks like that UV crosslinking in brain works less efficient as compared to the other tissues; please discuss why.

The signal sum of RBP identified by *ex vivo* eRIC is indeed lower in brain compared to liver and kidney. As discussed in the previous version of the manuscript, this is not due to differences in the abundance of RBPs (Fig 3b middle panel) or in the level, integrity and purity of the captured RNA (Figure 1b, Figure 1c, Supplementary Fig. 1a, Figure 3b bottom panel). It is also not due to biases in MS (pipeline related to Figure 1a), or to differences in UV-crosslinking efficiency (Figure 3d and Supplementary Fig. 2b). We rather think that the overall lower capture of RBPs in the brain is a result of the lower levels of poly(A) RNA in comparison

to kidney and liver (see the new Figure 3e of the revised manuscript and our answers to reviewers 1 and 3).

-Figure 3d: maybe label the figure with proximity labeling assay.

“Proximity labelling assay” was added as suggested by the reviewer.

-Figure 3c and supplementary figure 3b: the log₂ ratio is most likely referring to the ratio plus/minus UV?

We apologize if the meaning of the Log₂ ratio was not clear. To correct this, the following sentence has been added to the legends of Figure 3c and Supplementary Fig. 4b (former Supplementary Fig 3b): “The data are shown as the Log₂ ratio of protein abundance in each eRIC (or input) sample relative to the average protein abundance in corresponding eRIC (or input) sample”.

-Supplementary figure 1b and 3c: please catalogue the eRIC specific proteins (with the artificially imputed value) in a separate table.

The information requested by the referee was included in Supplementary Data 1. eRIC specific proteins can be sorted by setting the filter in the “Signal in input” column to FALSE.

-Please describe the “Hentze lab” SP3 workflow in more detail, e.g. which organic solvent was employed for bead protein aggregation (acetonitrile, ethanol, propanol)?

As requested by the reviewer, a detailed description of the SP3 workflow used in our study has been added in the methods section (see “Sample preparation for mass spectrometry (MS) and TMT labelling” in page 15).

-Instead of harnessing the total proteome, using the list of expressed genes/proteins (in a given organ; should be available from RNAseq data) is usually a much better “background proteome” for GO analyses.

We agree with the reviewer that the list of proteins actually present in the sample in principle represents the most appropriate reference for GO term enrichment analysis. However, using the proteins detected by MS for background correction is not free of bias: some proteins present in very small amounts in input samples may be below the detection limit of MS and be falsely omitted. This was actually the case for 158 (!) proteins that were detected in eRIC eluates but not in the inputs because of the enrichment by oligo(dT) capture. Nonetheless, we performed a new analysis, using the proteome of the input samples as reference. As shown in the tables provided below, we identified GO terms very similar to those of our previous analysis. These include RNA-related clusters such as “translation”, “regulation of RNA stability” and “mRNA splicing, via spliceosome” as well as metabolism-related clusters such as: “tricarboxylic acid cycle”, “fatty acid oxidation” and “NAD binding”. Thus, the method of background correction seems to have little effect on GO term enrichment in this case. Since the core conclusions of either analysis are very similar, we continue to use the original data, which we think are equally robust.

GO-BP	Brain		Kidney		Liver	
	Fold enrich.	adj P-val	Fold enrich.	adj P-val	Fold enrich.	adj P-val
translation	5.16750624	2.497E-47	2.1146344	9.0918E-25	4.36927754	1.64502E-71
cellular protein localization	2.506432656	3.796E-23			1.95449389	4.05788E-21
regulation of RNA stability	8.224340917	1.024E-18	3.2271529	5.06419E-13	6.4821722	2.81453E-26
cellular amino acid metabolic process	3.79020336	1.819E-08	1.67811951	3.39323E-06	5.27008613	1.05968E-36
mRNA splicing, via spliceosome	6.892719321	1.041E-29	1.89055256	4.41631E-06	4.24018408	7.44563E-24
nucleotide metabolic process	2.304204168	1.886E-05			2.43847057	4.57619E-13
Golgi organization	4.868322229	4.179E-06			3.28201499	1.76692E-05
vesicle localization	3.275763292	9.911E-08			2.12950904	6.08263E-05
lipid metabolic process					1.50316432	4.55195E-05
coenzyme metabolic process	2.426811594	0.001031	1.49203433	0.001174348	3.96428169	1.76566E-25
stress granule assembly	17.86133333	7.204E-11	3.14647408	0.003365161	11.1150908	2.23737E-11
endosomal transport	2.468814103	0.0021224			2.11664326	0.000300186
tricarboxylic acid cycle	9.367832168	6.367E-05	2.52559792	0.018115537	8.946806	2.7725E-10
lysosomal transport	3.543915344	0.0033579			2.48104704	0.009000627
fatty acid oxidation			1.74266257	0.01627197	4.98243054	3.56886E-13
protein localization to endoplasmic reti	5.423481781	0.000182	2.16409077	0.02653577	4.11331483	2.2326E-05
carbohydrate metabolic process	1.610096154	0.0827226	1.23561029	0.392490757	2.17091616	4.06239E-09
mRNA transport	5.221005128	6.559E-07	1.53441987	0.598136979	3.89028176	6.14763E-08
cytoskeleton organization	2.076899225	1.77E-09	1.11858901	0.764478192	1.69634379	6.50531E-09
glutathione metabolic process			1.51535875	0.813627007	3.26914434	0.001219517

GO-CC	Brain		Kidney		Liver	
	Fold enrich.	adj P-val	Fold enrich.	adj P-val	Fold enrich.	adj P-val
ribonucleoprotein complex	4.82476715	2.975E-58	2.96791974	1.99475E-21	4.06306722	1.8033E-84
cell projection	2.063789039	4.261E-18			1.68873733	6.99395E-17
cytoplasmic stress granule	14.21577763	1.022E-27	3.49675639	1.62201E-14	10.0860386	8.00397E-35
peroxisome					3.44772207	1.59387E-07
mitochondrial matrix	3.723125604	1.982E-10	1.82668394	8.52097E-06	5.17012536	6.20342E-44
nuclear envelope	2.512004853	1.557E-05			2.09203096	1.59121E-06
mitochondrial membrane	2.050811132	7.723E-05			2.58808943	3.64865E-19
proteasome complex	5.816663338	0.0001291			5.04919946	1.71404E-07
endoplasmic reticulum	1.588030828	0.0001686			1.79568076	1.9257E-14
P-body	5.992925863	6.089E-07	2.6348532	0.002417956	5.08241788	1.0556E-10
microtubule	3.353958509	2.165E-10	1.57322062	0.007224977	2.23435291	2.46083E-07
membrane raft	2.057068374	0.0066396			1.94354347	0.000137484
Golgi apparatus	1.710121523	4.203E-05			1.3117741	0.009317967
spliceosomal complex	6.075020738	1.196E-16	1.7166765	0.016018132	3.87821232	5.62214E-14
extracellular matrix			2.03936006	0.002115959	1.74717864	0.008649576
nuclear matrix	4.391762465	3.07E-05	1.86563679	0.020237538	3.86246515	2.55265E-08
endocytic vesicle	2.288208057	0.0351138			2.21778235	0.001161867
extracellular exosome	3.024467258	0.0386952	1.89940398	0.019682154	3.71549552	2.75823E-06
focal adhesion	2.360849582	0.0374521	1.49340638	0.126454506	3.19433671	3.49221E-08
actin cytoskeleton	2.374555531	4.81E-05	1.35309921	0.31403033	2.69357492	3.9246E-14
lamellipodium	3.360519176	6.954E-05	1.25760537	0.859052207	3.54660936	3.5151E-11

GO-MF	Brain		Kidney		Liver	
	Fold enrich.	adj P-val	Fold enrich.	adj P-val	Fold enrich.	adj P-val
mRNA binding	7.93947937	1.03E-51	2.5620126	3.63886E-24	5.9021326	5.22139E-64
RNA helicase activity	8.31261105	1.99E-10	3.0024271	1.93595E-07	6.6025686	4.16372E-15
ligase activity	5.09047681	3.36E-09	2.03309	5.76975E-07	5.4287786	9.94056E-23
lyase activity					2.5715267	3.00116E-05
translation initiation factor activity	7.8485897	2.66E-08	2.4524329	0.000417016	5.922304	1.0054E-10
nucleotide binding	2.15061221	2.8E-18	1.1973311	0.002463746	2.1564044	5.53677E-38
structural constituent of cytoskeleton	8.51985066	3.2E-11	2.1004634	0.005605439	4.9287596	1.43563E-08
cytoskeletal protein binding	2.61242683	4.52E-14	1.2816387	0.006404288	2.4093139	3.80108E-23
translation elongation factor activity	7.77010381	0.009807	3.2407149	0.004114413	7.8174413	1.99381E-06
NADP binding					3.3695867	0.008504734
acid-thiol ligase activity			2.4657614	0.019336073	6.9798583	7.53089E-06
heat shock protein binding	4.74839677	1.55E-07	1.5712557	0.125330877	3.5829939	1.21481E-08
C-acyltransferase activity			2.7005958	0.085159442	5.922304	0.003300831
AU-rich element binding	11.5626545	1.88E-06	2.520556	0.348392547	5.2348937	0.00277009
NAD binding	2.77503707	0.374769	2.1267192	0.002463746	6.2818724	7.42719E-13
flavin adenine dinucleotide binding			1.5495222	0.347576652	4.071584	3.11714E-07

Reviewer #3 (Remarks to the Author):

This manuscript from Perez-Perri, et al. describes a new method (“eRIC”) to identify proteins binding to poly(A)+ RNA in mouse organs. For a variety of technical reasons, including the inability for UV to penetrate tissues, this aspect of RNA biology has been unexplored, especially relative to our knowledge of proteins binding RNA in cultured cells. One strength of this manuscript is the development of this method (which involves tissue slicing and robust technical controls), which will also have applications to other UV-based methods. Another strength of this method are the associated datasets, which will be of interest to the RNA community. For the most part, this manuscript is clearly presented, and the interpretations are not overstated. The weakest part of the manuscript are the analyses on the differences between RBPs bound in the three different organs and in tissue culture cells. There are additional alternative interpretations, and the data (and discussion) can be a little hard to follow at times. Given that the following issues are addressed, I support publication of this study:

We are grateful to the referee for these positive, supportive comments.

1. For the new RBPs identified (i.e., those that have not shown up in previous RIC studies), how many are expressed in tissue culture cells? The types of analyses performed in Figure S4 would be useful for this class of RBPs as well.

We thank the reviewer for this judicious remark. Following this suggestion, we assessed whether the 291 novel RBPs identified by *ex vivo* eRIC were expressed in the cell lines used in published RIC datasets (the 30 datasets used in Figure 5 and Supplementary Fig. 7). This analysis is now included in the new version of the manuscript: remarkably, “263 out of the 291 novel RBPs (>90%) are expressed in at least one of ten tested cell lines (Figure 5d), with as many as 128 (~44%) expressed in at least half of them, and 42 (>14%) in all ten tested cell lines.”

2. The observation of altered binding of RBPs is very intriguing. However, this section would be stronger if the authors could also exclude some additional alternative explanations, such as reduced poly(A)+ RNA concentration in cells or altered protein localization.

We agree that organ-specific differences in the overall binding of RBPs to poly(A) RNA represents one of the interesting results of this study. Following the reviewer’s suggestion regarding the possible influence of poly(A) RNA levels, we quantified the poly(A) RNA content

of brain, kidney and liver tissues. Poly(A) RNA levels in the brain are indeed lower than in the kidney and liver. The apparently low poly(A) RNA-binding activity in the brain could thus result from the relatively lower amount of poly(A) RNA in this organ. However, this consideration would not apply to comparing kidney and liver, since the overall association of proteins with poly(A) RNA is lower in the liver than in the kidney, although the liver contains equal or even higher amounts of poly(A) RNA. It is well established that numerous RBPs can interact with both poly(A) and non-poly(A) RNA (PMID: 27018577, PMID: 32252787, ENCODE project). We considered the possibility that the proportion of RBPs bound to poly(A) RNA or non-poly(A) RNA might be affected by the stoichiometry of the two RNA biotypes. To address this experimentally, we first measured non-poly(A) RNA levels in brain, kidney and liver and found they are higher in the liver than in the kidney and lowest in the brain (Figure 3e). We then isolated and identified the RBPs bound to non-poly(A) RNA in each organ, and combined the non-poly(A) RBP data with our previous eRIC data on poly(A) RBP to determine the relative binding of proteins to poly(A) and non-poly(A) RNA. We detected 222 RBPs that interact with both poly(A) and non-poly(A) RNA (dual binders). Interestingly, dual binders show decreased interaction with poly(A) RNA in liver relative to kidney. However, this is apparently not the case for proteins that interact exclusively with poly(A) RNA. Possibly, the higher amount of non-poly(A) RNA in liver “competes” with the binding of RBPs to poly(A) RNA. Together, our results suggest that the overall poly(A) RNA content, as well as the relative levels of poly(A) versus non-poly(A) RNA may impact on the composition of the poly(A) RNA interactome. These new data are presented in Figures 3 and 4 and discussed on pages 4-5 and 10-11 of the manuscript.

Note: The link between poly(A) RNA levels and overall RBP activity does not seem to be a feature limited to tissues, but has also been observed in cultured cells. For example, we found that Jurkat cells (immortalized human T lymphocyte cell line) contain more poly(A) RNA than human primary T cells from healthy donors. We identified about 800 RBPs in Jurkat cells, but only 50 to 100 RBPs in primary lymphocytes, although similar levels of captured poly(A) RNA were used as input (Perez-Perri et al., unpublished). We have not included or discussed these results in the present manuscript, because they are purely correlative at this point and require further in-depth analysis.

The reviewer also mentions the potential influence of protein localization on RBP activity. While this is an interesting possibility, it cannot readily be addressed by *ex vivo* eRIC given that we processed whole tissue lysates. Nonetheless, a recent study from our lab (Backlund et al., NAR, 2020) has reported localization-dependent activity in cultured cells.

3. I was a little confused by the statement that many (how many?) RBPs did have different expression. A clearer analysis of the data would really help the reader.

We assume the reviewer refers to the statement “This suggests that differential RBP binding to RNA across mouse organs most commonly results from differential RBP expression. This standard pattern has however numerous exceptions, and dozens of RBPs exhibit differential RNA binding without commensurate changes in overall protein abundance (new Supplementary Fig. 4c, Supplementary Data 5)”.

For clarification, we modified Supplementary Figure 4c to specify the number of proteins within each category (i.e. hit, no hit, candidate, etc.). We also inserted three additional data tables (worksheets “Brain vs Kidney”, “Brain vs Liver” and “Liver vs Kidney”) into Supplementary Data 5 that show pairwise comparisons of protein intensities between organs, either in the eRIC eluates or the input samples. These data tables can be filtered to identify proteins with a given intensity pattern, for example proteins with similar expression in two organs but displaying stronger RNA association in one or the other organ. In addition, these tables depict the color of each protein in Supplementary Fig. 4c. We hope that these changes clarify the understanding of Supplementary Fig. 4 and Supplementary Data 5.

Minor comments:

1. P1, line 49: “only exceptionally promotes protein-protein crosslinking” is unclear.

The phrase has been replaced, it now reads as follows: “Irradiation of cells with ultraviolet (UV) light has been widely used to crosslink single-stranded nucleic acids and proteins, because UV light is inefficient in protein-protein crosslinking; hence it selects for direct RNA-protein interactions”.

2. Some of the word usage is awkward, and some additional editing would help.

We carefully revised the manuscript and improved the wording.

REVIEWER COMMENTS

Reviewer #1 (Remarks to the Author):

The manuscript by Perez-Perri and colleagues has certainly been improved with the new experiments added, including the non-poly(A) RNA-bound proteomes which are indeed interesting and open new research avenues. As I said in my previous revision is a very classy and useful resource, but in my opinion, I think this article still lacks the biological relevance. I understand that performing CLIP in tissues would be challenging, but I consider that having atlases without biological meaning decreases the impact of the study as a scientific article. Indeed, in the rebuttal letter, the authors state that the aim of this paper is to 'establish a technique that empowers researchers to determine the RNA-bound proteomes of intact organs under different conditions' and 'provide the first comprehensive RBP atlases from mammalian organs' so agreeing with the second reviewer I would suggest a methods journal.

Comment #1:

I thank the authors for their reply. They have fully addressed my comment about the possibility that the UV irradiation would may promote protein-protein interactions as previously reported by other authors.

The experiments are very nicely designed and the UV titration is very illustrative.

Comment #2:

I thank the authors for their reply. I agree with them that this article extends the state of the art, but from that perspective, every new atlas of RBPs in different organisms, cell type, organ, etc... will extend the state of the art. I still miss the biological relevance, what is the relevance? What is the function of the RBPs exclusively detected in one of the organs? Do they have any relevant function for the organ? The new experiments of the non-poly(A) RNA-bound proteomes are interesting and of course add impact to the article, but again are lists of proteins with no biological relevance. For me, the most interesting finding in this regard is the observation that the overall poly(A) RNA content, as well as the relative levels of poly(A) versus non-poly(A) RNA may impact on the composition of the poly(A) RNA interactome.

So, overall, although they do not address my comment about the biological relevance, it is true that new experiments are very well performed and provide novel information not reported so far.

Reviewer #2 (Remarks to the Author):

Review revised manuscript

The revised manuscript of Perez-Perri and colleagues has substantially improved the impact of the work and resolved many of the concerns raised by all three referees.

Still, the work is mainly descriptive and does not provide any biological/physiological data (functional, mechanistic etc.). Nevertheless, the additional data and experiments allow us to come to two main conclusions. First, the authors clearly demonstrate that *ex vivo* eRIC and hence probably other UV-crosslinking mediated RBPomes are technically highly specific. Hence, the skepticism regarding a potential huge number of false positive RNA binding proteins “contaminating” the RBP catalogues (mainly due to presence of rRNA) can be put to rest. Second, the additional extensive studies carried out on non-poly(A) binding RBPs suggest that the observed tissue-/organ-specific differences in both RBPs and RNA binding activities are likely biological meaningful! In this context, the description of dual mode binders is quite interesting. In detail, the Hentze laboratory can now nicely show that the RBP activity of the majority of enzyme-RNA interactions represents a more ‘general’ RNA binding activity and is not selective for poly(A) RNA (mainly mRNA). This might hint to a “rheostat” function of RNA with ample of regulatory opportunities governed by putative competition and squelching mechanisms. Overall, in its current form the manuscript classifies as a method paper of very high quality. For now, *ex vivo* eRIC will probably be exclusively adopted by specialized laboratories but the work will hopefully boost further work in this direction. In case the editorial board is not concerned too much in regard to the limitations in its biological impact, I would like to support the publication of the revised manuscript in Nature Communications.

There are just a few minor issues that I have addressed in my answer to the rebuttal letter (which do not require any further experiments or analyses).

Additional minor comments are listed below.

Fig. 1d (also concerns other Volcano plot figures): The number of biological replicates is 4. They have been pooled to N=2 (by the way, have you utilized some tool like “voomWithQualityWeights” in order to increase the weight for the pooling effect? Like in doi: 10.1093/nar/gkv412;). Thus, the statistics is based on the simultaneous data analysis employing –UV (N=4) and +UV (N=8) data groups from kidney, liver and brain (altogether). Is this correct? Please clearly indicate that N=2 for +UV (per tissue) and N=1 for –UV (per tissue). It would be also nice to update Fig. 1d so that it is

congruent with the novel data (e.g. presented in Fig. 4b), thereby indicating the “candidate” proteins with ‘blue dots’.

The experiments mainly presented in Suppl Fig. 5 and 6 are quite sound. I agree with the authors that these data completely rule out a prominent role of UV-mediated protein-protein crosslinking. The spread of protein separation in SDS-PAGE (sometimes pure Gaussian or Gaussian with a tail towards the lower molecular weight) is known and the results are expected. The effects seen in the high molecular weight range are due to aggregation of proteins (partially getting stuck in the well and forming a “smear”). The bimodal distribution of Pkm, Eno1 and Gapdh represents an interesting observation but we have seen similar things with proteins not binding RNA. Sometimes, this is reformation of disulfide bridges (blocking the cysteine residues PRIOR to SDS-PAGE with iodoacetamide might resolve it). In other instances, there are proteins that still adopt some residual secondary structure in SDS-PAGE or are still interacting with other biomolecules under SDS-PAGE conditions.

Line 643: {Asenico, 2018....} – reference to bibliography is broken

Line 1025, 1026: formatting of reference!

Reviewer #3 (Remarks to the Author):

This revised manuscript from Perez-Perri and colleagues represents a substantial improvement from the initial submission. I commend the authors for the additional experiments, especially the analysis of “piggy-back” crosslinking, poly(A) vs non-poly(A) binding, and the expression of new RBPs. The additional experiments and analysis have significantly improved the study, and I support publication in its current form.

REVIEWER COMMENTS

Current replies by the authors shown in green.

Previous replies by the authors shown in blue.

Reviewer #1 (Remarks to the Author):

The manuscript by Perez-Perri and colleagues has certainly been improved with the new experiments added, including the non-poly(A) RNA-bound proteomes which are indeed interesting and open new research avenues. As I said in my previous revision is a very classy and useful resource, but in my opinion, I think this article still lacks the biological relevance. I understand that performing CLIP in tissues would be challenging, but I consider that having atlases without biological meaning decreases the impact of the study as a scientific article. Indeed, in the rebuttal letter, the authors state that the aim of this paper is to ‘establish a technique that empowers researchers to determine the RNA-bound proteomes of intact organs under different conditions’ and ‘provide the first comprehensive RBP atlases from mammalian organs’ so agreeing with the second reviewer I would suggest a methods journal.

We thank the reviewer for the appreciation of the method described in our work. We agree it opens new research avenues. We appreciate that there are several biological questions that remain to be tackled, and hope that the methodology described here will provide a powerful mean to do so. Nonetheless, we hope that the work is not purely technical and does not represent ‘atlases without biological meaning’. In addition to other biologically important leads, the demonstration that a body of work published by numerous groups on the basis of cultured cells has an in vivo (ex vivo) counterpart is in itself biologically meaningful.

Comment #1:

I thank the authors for their reply. They have fully addressed my comment about the possibility that the UV irradiation would may promote protein-protein interactions as previously reported by other authors.

The experiments are very nicely designed and the UV titration is very illustrative.

We thank the reviewer for the appreciation of the extensive experimental work added to the manuscript. Based on the reviewers’ earlier comments, we decided that a substantial investment of time and resource was warranted to settle doubts that may also have remained with a fraction of the readership, and we look forward to sharing the new results in the published manuscript.

Comment #2:

I thank the authors for their reply. I agree with them that this article extends the state of the art, but from that perspective, every new atlas of RBPs in different organisms, cell type, organ, etc... will extend the state of the art. I still miss the biological relevance, what is the relevance? What is the function of the RBPs exclusively detected in one of the organs? Do they have any relevant function for the organ? The new experiments of the non-poly(A) RNA-bound proteomes are interesting and of course add impact to the article, but again are lists of proteins with no biological relevance. For me, the most interesting finding in this regard is the observation that the overall poly(A) RNA content, as well as the relative levels of poly(A) versus non-poly(A) RNA may impact on the composition of the poly(A) RNA interactome.

So, overall, although they do not address my comment about the biological relevance, it is true that new experiments are very well performed and provide novel information not reported so far.

We agree that the potential global regulation of protein-poly(A) RNA interactions by the overall content of poly(A) RNA or by the levels of poly(A) relative to non-poly(A) RNA levels is an interesting biological concept that emerges from this manuscript. In addition, we think this study extends the state of the art in a meaningful way by providing a method that holds promise to open new research opportunities, for example using animal models and clinical specimens.

Reviewer #2 (Remarks to the Author):

Review revised manuscript

The revised manuscript of Perez-Perri and colleagues has substantially improved the impact of the work and resolved many of the concerns raised by all three referees. Still, the work is mainly descriptive and does not provide any biological/physiological data (functional, mechanistic etc.). Nevertheless, the additional data and experiments allow us to come to two main conclusions. First, the authors clearly demonstrate that ex vivo eRIC and hence probably other UV-crosslinking mediated RBPomes are technically highly specific. Hence, the skepticism regarding a potential huge number of false positive RNA binding proteins “contaminating” the RBP catalogues (mainly due to presence of rRNA) can be put to rest. Second, the additional extensive studies carried out on non-poly(A) binding RBPs suggest that the observed tissue-/organ-specific differences in both RBPs and RNA binding activities are likely biological meaningful! In this context, the description of dual mode binders is quite interesting. In detail, the Hentze laboratory can now nicely show that the RBP activity of the majority of enzyme-RNA interactions represents a more ‘general’ RNA binding activity and is not selective for poly(A) RNA (mainly mRNA). This might hint to a “rheostat” function of RNA with ample of regulatory opportunities governed by putative competition and squelching mechanisms. Overall, in its current form the manuscript classifies as a method paper of very high quality. For now, ex vivo eRIC will probably be exclusively adopted by specialized laboratories but the work will hopefully boost further work in this direction. In case the editorial board is not concerned too much in regard to the limitations in its biological impact, I would like to support the publication of the revised manuscript in Nature Communications.

There are just a few minor issues that I have addressed in my answer to the rebuttal letter (which do not require any further experiments or analyses).

We thank the reviewer for the positive appreciation of the quality and value of the additional experimental work, and for recommending publication in Nature Communications.

Additional minor comments are listed below.

Fig. 1d (also concerns other Volcano plot figures): The number of biological replicates is 4. They have been pooled to N=2 (by the way, have you utilized some tool like “voomWithQualityWeights” in order to increase the weight for the pooling effect? Like in doi: 10.1093/nar/gkv412;).

We did not treat the eRIC eluates obtained from independent mice that were pooled before the MS analysis as independent biological replicates. Thus we did not use any tool like “vroomWithQualityWeights” to increase the weight of the pooling effect.

Thus, the statistics is based on the simultaneous data analysis employing –UV (N=4) and +UV (N=8) data groups from kidney, liver and brain (altogether). Is this correct?

The statistics are based on the integrated data analysis of brain, kidney and liver samples (altogether) employing, per organ: -UV (N=1, pool of the respective organs of 4 mice) and +UV (N=2, see next response).

Please clearly indicate that N=2 for +UV (per tissue) and N=1 for –UV (per tissue).

The following sentences were added to the legends of Figures 1e and d:
“for each organ, four +UV eRIC eluates were generated, each derived from a single mouse; eRIC eluates from two mice were combined, rendering n=2. Organ sections from four mice were pooled to generate one -UV eRIC eluate per organ (n=1).”

It would be also nice to update Fig. 1d so that it is congruent with the novel data (e.g. presented in Fig. 4b), thereby indicating the “candidate” proteins with ‘blue dots’.

Candidate proteins have been indicated with blue dots in Figure 1d.

The experiments mainly presented in Suppl Fig. 5 and 6 are quite sound. I agree with the authors that these data completely rule out a prominent role of UV-mediated protein-protein crosslinking. The spread of protein separation in SDS-PAGE (sometimes pure Gaussian or Gaussian with a tail towards the lower molecular weight) is known and the results are expected. The effects seen in the high molecular weight range are due to aggregation of proteins (partially getting stuck in the well and forming a “smear”). The bimodal distribution of Pkm, Eno1 and Gapdh represents an interesting observation but we have seen similar things with proteins not binding RNA. Sometimes, this is reformation of disulfide bridges (blocking the cysteine residues PRIOR to SDS-PAGE with iodoacetamide might resolve it). In other instances, there are proteins that still adopt some residual secondary structure in SDS-PAGE or are still interacting with other biomolecules under SDS-PAGE conditions.

We thank the reviewer for the thorough observations and for sharing their experience. We agree with the interpretations.

Line 643: {Asenico, 2018....} – reference to bibliography is broken
Line 1025, 1026: formatting of reference!

We thank the reviewer for these meticulous observations. Both problems have been fixed in the current version of the manuscript.

Reviewer #2 (Remarks to the Author): (From attachment)

General overview

The provided manuscript by Hentze, Galy and colleagues describes a proteomics methodology that enables the UV crosslinking based interrogation of RNA binding proteins (RBPs) in mouse organs *ex vivo* at proteome scale (RBPome). Their workflow builds on their previously published eRIC technology, which utilizes LNA containing capture probes (targeting polyA RNAs) and is deemed to provide improved specificity and efficiency. In order to overcome the problem that penetration depth of UV light in mammalian organs is quite limited, the authors adopt and streamline an established technology from the field of electron microscopy namely cryosectioning of tissues. As a test case, they embarked on studying murine brain, liver and kidney, thereby significantly expanding the current list of mammalian RBP candidates identified by mass spectrometry in mammalian organs (towards this end there has been only one global approach published that employed formaldehyde (FA) crosslinking of mouse tissue: FAX-RIC, NAR 49, 2021). Technologically, this is an important step forward for the community. The proposed work is with no doubt a classical high quality piece of research (for which the Hentze laboratory is well known for) and is already written in a quite elegant way. However, after reading it my main conclusion is: “Yes, we know it!” For me the story does not trigger a single “heureka moment” and even worse, the oeuvre does not provide any real biological insight! Therefore, I would advise publishing the manuscript in a more specialized or methods focused journal (like Molecular Systems Biology, Scientific Reports or npj systems biology and applications). A more detailed argumentation is outlined in the following.

We thank the reviewer for the positive comments on the quality of our work. As detailed above in our responses to reviewer #1, we believe that this work makes substantial contributions that advance the state-of-the-art in ways that qualify it for a broad interest journal. Mechanistic functional data are lacking, because they would “explode” the timeframe and scope of a single publication.

My pleasure! Finally, I agree – would be too much of extra work!

One of the major claims of Perez-Perri et al. is the discovery of enzymes from intermediary metabolism as pervasive RNA binders in the three mouse organs (under investigation). Until now, the dozens of UV- or FA-assisted global RBPome catalogues have identified a hundred of metabolic enzymes and the question remains if most of them are just bystanders (for instance RNA has a general role in preventing protein aggregation: EMBO Reports (2020)21:e49585) or really play a causative functional biological role. The finding of Perez-Perri et al. (figure 6) that especially enzymes harboring nucleotide binding domains represent a substantial fraction of their RNA interacting enzyme catalogue may point towards this direction. Of note, it has been proposed by Tony Hyman that the cellular ATP concentration is critical for regulating protein phase separation, solubility and aggregation. Likewise, other nucleotide-containing cofactors or RNA might elicit similar effects.

We agree with the reviewer that RNA-protein interactions can play different roles and that more are emerging. For example, we recently reported that the enzymatic activity of enolase 1 is directly controlled by RNA and that this riboregulation is critically relevant for mouse embryonic stem cell differentiation (Huppertz et al., Mol. Cell, 2022). The candidate RBPs identified by *ex vivo* eRIC could therefore have different RNA-binding

functions. Modulation of protein aggregation could well be one of them, and we previously reported that nearly half of the RNA-binding regions of RBPs fall into disordered domains (Castello et al., Mol Cell, 2016).

Agreed! Nonetheless, this manuscript does not add any functional or mechanistic explanation.

In addition, I do not even want to discuss the potential issues (identifying abundant metabolic enzymes) associated with substantial rRNA contamination in RBPome captures (also seen in this study: figure 1b and suppl. figure 1a)!

We previously reported that rRNA accounts for about 3% of the RNA captured by eRIC as opposed to RIC (PMID: 30352994). We do not consider this substantial, especially since rRNA represents ~ 90% of cellular transcripts. Previous attempts to completely remove rRNA from eRIC eluates have been associated with concomitant and proportional loss of poly(A) RNA (PMID: 30352994). The reasons for the presence of rRNA in eRIC samples remain unclear. The rRNA could be co-isolated due to RNA-RNA interactions with poly(A) transcripts. The addition of poly(A)-rich tails to the 3' end of rRNA, as observed in yeast and human cells (PMID: 15173578, PMID: 16738135, PMID: 20368444), could also mediate specific (!) binding

to the oligo(dT) probe. Data added in revision and discussed above (responses to reviewer #1) further address the issue raised by the reviewer.

Regarding the role of protein abundance on eRIC data, we show that hundreds of proteins that are very abundant in the inputs are not detected in eRIC eluates (Supplementary Fig. 1b). In addition, we have now characterized the non-poly(A) RNA interactome of organs (see Figure 4, Supplementary Fig. 3 and section "Organ-specific interactions of RBPs with poly(A) and non-poly(A) RNA" on pages 4 and 5 of the revised manuscript), and found that many ribosomal proteins (51), which represent some of the most abundant RBPs, are expectedly identified as non-poly(A) binders, but are not detected in eRIC eluates. While we do not exclude that a few particularly abundant proteins may be false positives, our results overall indicate that ex vivo eRIC captures RBPs and more particularly poly(A) RBPs with high specificity.

I am very thankful to the authors to carry out this substantial amount of extra work. The data concerning the non-poly(A) binders are very interesting and provide a valuable source of information. Together, with eRIC captures of poly(A) associated RBPs this study (maybe for the first time) demonstrates that at the level of protein detection by LC-MS, RNA contaminations originating from highly abundant cellular RNAs (like rRNA) are virtually irrelevant! Or in other words, compared to other OMICs approaches the small amount of FPs can be tolerated.

The authors would also like to suggest that the literature already demonstrates/proves the widespread specific biological function of enzyme RNA interactions but the given citations (#35, #37, #38, #39, #44, #45, #46) reveal that most of this insight was gained by hypothesis-driven

approaches, like in vitro assays (EMSA, FA-XL RNA-IP), ribosome-polysome analysis, RNA aptamer affinity pull-downs, or the much more specific and sophisticated RBDmap methodology (pioneered by the Hentze laboratory).

As mentioned by the referee, the biological roles of several enzyme-RNA interactions have

been uncovered using entry points different from RIC. Unlike more historical methods, RIC as has only been available for about 10 years. The critical point is that RIC and especially eRIC experiments (re-)identify enzyme RBPs for which biological roles are clearly established. This also includes enzymes such as ENO1, SAHH, PKM2, or FASTKD2, for which the biological role of RNA binding has only recently been determined.

Agreed! Hopefully, the RNA biology community will further address these observations in follow-up investigations.

Moreover, the fact that mammalian tissue RBPomes will be different compared to standard cell culture RBPomes (HeLa, HEK293, mESCs etc.) is also not very surprising and similar observations were made in the context of multiprotein complexes (e.g. the groundbreaking work of A. Ori and M. Beck).

Our starting hypothesis was that organ and cell culture RBPomes would largely overlap, with possibly some systematic differences that would be rather difficult to guess. Our experiments therefore did not aim to uncover surprises, but to experimentally determine the facts. Ex vivo eRIC allowed for the first time the system-wide comparison of RBPomes from organs and cultured cells. We agree with the reviewer that differences between the RBPome of tissues and the RBPome of cultured cells are to be expected. Although technical aspects also have to be considered, we think that at least some of the differences between the RBPomes of organs and cultured cells represent interesting biological observations. They may, for example, reflect the influence of the metabolic environment or oxygen tension, or be attributable to the often cancerous origin and high proliferation rate of cultured cells, whereas intact tissues are non-transformed and essentially quiescent. We thus provide experimental datasets to explore the role of these parameters on RNA-protein interactions in the future. In the revised version of the manuscript we have assessed whether the novel RBPs identified in organs (Figure 5a-c) are at all expressed in the cell lines used in previous RBP profiling studies. Remarkably, we have found that 263 out of the 291 (>90%) novel RBPs are expressed in at least one of ten tested cell lines, with 42 RBPs (>14%) expressed in all of them (Figure 5d).

Thank you. These additional experiments more or less resolve the issue.

Furthermore, because mammalian organ UV-XL RBPomics is technically challenging, the depth of analysis is severely hampered as compared to studies carried out in cellulo. Hence, the comparison of the organ RBPomes with the corresponding cellular data is of very limited value, because it faces a kind of “apple and oranges comparison problem” (figure 4b, figure 5a, supplementary figure 4a, b, c).

UV crosslinking of RNA-protein interactions in intact tissues is indeed non-trivial, and no in-depth

RIC data from mammalian organs were reported since the in cellulo work by Castello et al., and Baltz et al. about a decade ago. Ex vivo eRIC alleviates the bottleneck of low UV penetration into tissues and has allowed the identification of about 1300 RBPs. This number is perfectly in line with the yield of typical eRIC studies in cultured cells. Ex vivo eRIC not only

detects most of the canonical RBPs that make up the core of the mammalian cell RBPome, but also identifies previously unknown RBPs. We think that this shows that the depth of mammalian organ RBPomics is fully in line with the earlier cultured cell studies, and hence

argues against them being “severely hampered”. As discussed above, we also think that comparisons between organ and cell-based RBPomes have been perfectly possible and have yielded informative first results.

I fully admit that the *ex vivo* eRIC experiments conducted with mouse organs are technically challenging and time consuming. Unfortunately, this will also restrict the utilization to highly specialized laboratories.

However, I have to congratulate the authors, because it was very smart to choose brain as one of the organs since neurons are evolutionary well known for their widespread and diverse use of RNA-based regulatory processes (e.g. alternative splicing, RNA editing, extended 5'UTR and 3'UTR expansion in neuronal mRNAs; thus ample of opportunity to catch novel RBPs). Unfortunately, in brain glia cells are at least as abundant as neurons and many of these neuronal RNA processes take place in a quite localized manner (axons etc.). Therefore, it will be very difficult to capture them by the current sensitivity of the presented technology (only two brain-specific candidate RBPs are discovered, figure 3a; but at least the GO cellular component AU-rich element binding is enriched, figure 2b).

We agree with these considerations. Our *ex vivo* eRIC study performed with whole tissue samples is a first step towards the exploration of the RBPome of intact organs, including the brain. Further refinement of the method may help studying the RBPome of specific tissue substructures in the future, especially also with the ongoing refinements in sensitive protein identification and quantification by mass spectrometry.

Yes, a lot of streamlining required for making it widely accessible.

So as to gain more functional biological insight the Hentze team could attempt several experiments. Scale up the study, thereby increasing the depth of analysis, which would allow them to conduct RBDmap. The latter will uncover the RNA binding domains or regions of the RNA interacting metabolic enzymes and presumably disclose some common or general binding/regulatory mechanism. Alternatively, an iCLIP or similar type of protein-centric methodology selecting a couple of RBP candidate metabolic enzymes might give a deeper biological insight based on the type and nature of bound RNAs. Another option would be to team up with metabolomics experts and scrutinize a possible correlation between the different organ metabolomes (using a cell culture metabolome as control) and the type of RNA interacting metabolic enzymes.

We thank the reviewer for these relevant suggestions. Conducting RBDmap in tissues, defining the RNA interactome of specific enzyme-RBPs using CLIP, or looking for potential correlations between the metabolome and RBPome of different organs could and likely will provide additional information (as well as complexity). However, we think that these extensions are beyond the scope of the current study. Therefore, we have focused our additional experimental work on the validation of the method and reported datasets rather than on broadening the scope of the work. Nevertheless, we agree that these studies represent interesting avenues for future research.

Based on the additional results presented in the revised version of the manuscript I would guess that with the current LC-MS setup employed by the authors it will be impossible to carry out RBDmap experiments except one would drastically increase the number of animals etc. But it is a pity that the authors did not have a look at the

metabolomics data available for murine tissues, especially in light of the fact that the term ‘metabolism’ plays such a prominent role in the manuscript.

Interestingly, we had previously considered to explore potential correlations between the RNA-binding activity of enzyme-RBPs across organs and published organ metabolomes. However, we had to realise that all metabolomics studies including mouse brain, kidney and liver tissues that we found were generated using experimental conditions (mouse feed, age, germ-free mice, etc) profoundly different from the conditions used in our studies. Quantitative cross-comparison between any of these datasets and our *ex vivo* eRIC data would thus lack a solid scientific basis. Following the reviewer’s comment, we recently found a publication (Sugimoto M. et al., Nucleic Acids Res. 2012, PMID: 22139941) describing metabolomics data of mouse brain, kidney and liver generated under conditions that are much closer to ours. Upon request, the authors of this study kindly shared their primary datasets with us. Unfortunately, these provide quantitative information for only ~50 metabolites out of the 376 metabolites linked to the 250 enzyme-RBPs described in Supplementary Data 7 of our study. Therefore, the suggested analysis cannot be performed rigorously with the data currently available.

As an alternative, we searched for correlations between the relative RNA-binding activities of organ enzymes and cofactor usage. This search did not detect any statistically significant enrichments (Fisher's exact test) for cofactor usage in any of the enzyme-RBP categories highlighted in Supplementary Fig. 4c and Supplementary Data 5. This result suggests that differences between the relative levels of cofactors across brain, kidney and liver do not represent primary control points for the interaction of enzymes with RNA. That said, we identified some enzyme-RBPs that display differential RNA binding across organs that cannot be accounted by differences in protein expression levels or by the global, organ-specific control of RBP activity (as described in Figure 3 b, c). These enzyme RBPs represent possible candidates for further exploration of potential organ-specific control of RNA binding by metabolites.

The graphics below display pairwise comparisons across brain, kidney and liver of the Log₂ fold change of eRIC signal normalized by the Log₂ fold change of input signal of enzyme-RBPs (Supplementary Data 7) depicted according to the cofactors employed by the enzymes. Names of enzymes with Log₂ values >1.5 or <-1.5 are displayed. Due to the lack of significant trends, these graphics are included here but not in the manuscript (primary data is provided in Supplementary Data 7).

Major points:

-It is not really clear to me how many (technical) replicates were measured by nLC-MS? The authors admit that only two biological replicates of UV-irradiated mice were employed (each of them representing a pool of two individual mice) in conjunction with a single pool of four non-UV control mice. I guess that many cryosection specimens can be created out of one organ? Hence, a more detailed description of the data analyses and statistical validation workflows is required.

As requested by the referee, a more detailed description of the protocol and data analysis has

been included in results and the materials and methods sections of the revised manuscript (see sections “Refinement of eRIC to characterize the poly(A) RNA-bound proteomes of mammalian organs”, and “MS data analysis” in pages 2 and 16, respectively). Regarding the amount of material needed for ex vivo eRIC, 30 µm cryosections are placed on a glass slide and cover an area of approximately 7 cm² (~60% of the area of a 4.6x2.5 cm glass slide). For the brain, this represents about 15-20 tissue sections (in the horizontal plane) per slide. For liver, this corresponds to 10-15 slices; for kidney, 20-30 sections. We determined that deep RBPomes can be obtained using about 30 glass slides per sample, corresponding to a surface of about 200 cm² of 30 µm thick tissue sections (equivalent to 1.5 brains, 2/3rd of the left lateral lobe of a liver, and 3 kidneys, respectively).

Thanks a lot. This is valuable information for the reader!

-Figure 1a, right panel and methods: how is the total protein normalization achieved in more detail? By measuring an aliquot of each sample before mixing or by mixing aliquots of the samples with different mixing ratios?

To compare RBPomes from different organs, normalization prior to MS analysis was performed on the basis of the amount of RNA captured, as in previous RIC studies. We used the amount of captured RNA for normalization, because it can be precisely measured (the mass of RNA that was submitted for each sample is shown in Figure 3b). By contrast, the determination of protein content is less accurate, because the absolute mass of captured protein is very low and typically below the detection limit of standard protein quantification methods (e.g. Bradford, BCA, etc.). Performing a pre-MS analysis of a small aliquot (<10%) of each sample to estimate the protein content, and combining samples in a mixing ratio to obtain the same amount of protein per sample would represent an alternative.

All right. RNA normalization is fine in case UV crosslinking efficiency (and RNPcomplex solubilization) amongst the different tissues will be similar. Although, the authors have tried to further address this problem it is still not fully clear to me if there is still an impact of these technicalities or not. Admittedly, it is difficult to tackle the problem experimentally!

-Is it possible to determine the total protein and RNA content of individual organ cryosection specimens?

The following numbers were extrapolated from the amount of cryosections used and the quantity of RNA and protein present in pooled samples:

Brain: 41.6 ng of poly(A) RNA per section, 342 µg of protein per section.

Kidney: 35.5 ng of poly(A) RNA per section, 94.4 µg of protein per section.

Liver: 113.4 ng of poly(A) RNA per section, 304.2 µg of protein per section.

We have not included this information in our revised manuscript, but we would be happy to do so, if requested.

Please include this information in the manuscript because it will allow the reader to decide whether to embark on ex vivo eRIC experiments or not!

The following sentence has been introduced to the section “eRIC: Cryosectioning, UV irradiation and cell lysis”: “Each section contains the following approximate quantity of poly(A) RNA and total protein (numbers extrapolated from the amount of cryosections used

and the quantity of RNA and protein present in eRIC eluates and pooled samples, respectively): brain: 41.6 ng of poly(A) RNA and 342 µg of protein; kidney: 35.5 ng of poly(A) RNA and 94.4 µg of protein; liver: 113.4 ng of poly(A) RNA and 304.2 µg of protein.”

-Figure 1b and supplementary figure 1a: it is highly advisable to characterize the RNAs by deep RNAseq (possibly with and without “ribodepletion”), which became an easy, straightforward and cheap QC technology that will even reveal the content of polyA harboring lncRNAs.

We estimated the levels of RNA captured qualitatively by capillary electrophoresis (Supplementary Fig. 1a) and quantitatively by qPCR (Fig. 1b). While we agree that deep RNAseq could be performed on the samples, we do not think that this would add to the data essential information, although one may well consider doing so in future studies. Since ex vivo eRIC isolates polyadenylated RNAs in an unbiased way, it is fair to expect that polyadenylated lncRNAs will contribute to the overall RBPomes.

Yes, probably too much of work for this manuscript. But would maybe shed light on the differences in RNA binding activities between different organs.

Minor points:

-Discuss mRNA stability/half live in brain, kidney and liver.

To our knowledge, no system-wide studies of RNA turnover rates in mouse brain, kidney, and liver have been performed. We report RBPs bound to the steady state transcriptomes within the respective tissues. These include RBPs previously implicated in the control of RNA stability/half lives.

Thanks!

-Figure 1e: please add the same scatter plots exclusively considering RBP harboring proteins. Figure 1e shows the normalized signal sum of proteins present in eRIC eluates and thus crosslinked to RNA. The scatter plots already present RBP data only. Or did the reviewer mean RNA-binding domain (RBD)-harboring proteins?

My apologies. I indeed meant RBD-harboring proteins.

We highlighted the proteins containing a known RBD in Figure 1e.

-Figure 3b: it looks like that UV crosslinking in brain works less efficient as compared to the other tissues; please discuss why.

The signal sum of RBP identified by ex vivo eRIC is indeed lower in brain compared to liver and kidney. As discussed in the previous version of the manuscript, this is not due to differences in the abundance of RBPs (Fig 3b middle panel) or in the level, integrity and purity of the captured RNA (Figure 1b, Figure 1c, Supplementary Fig. 1a, Figure 3b bottom panel). It is also not due to biases in MS (pipeline related to Figure 1a), or to differences in UVcrosslinking efficiency (Figure 3d and Supplementary Fig. 2b). We rather think that the overall lower capture of RBPs in the brain is a result of the lower levels of poly(A) RNA in comparison

to kidney and liver (see the new Figure 3e of the revised manuscript and our answers to reviewers 1 and 3).

OK. But this leaves one confusion with the y-axis in Suppl. Fig. 2b. The signal sum is given as [10e-6]. This would be noise in LC-MS except these values are normalized by large integer numbers?

We thank the reviewer for this careful observation. The signal sum in Supplementary Fig 2b corresponds in fact to large and not small numbers, but we have miscommunicated this point. “x 10e-6” in the y-axis of the previous version intended to represent that the original signal sum multiplied by 10e-6 equals the value provided in the figure (e.g. a value of 100 in the y-axis represents a signal sum of 100×10^6 , because $100 \times 10^6 \times 10^{-6} = 100$). The label of the y-axis has been changed to: “x 10e6” in the new version of the manuscript.

In detail, the PLA data kind of show that kidney behaves as a quite heterogenic tissue as recently shown: Wang, G. et al., *Nature Metabolism* volume 4, pages 1109–1118 (2022); <https://doi.org/10.1038/s42255-022-00615-8>.

We thank the reviewer for this information.

-Figure 3d: maybe label the figure with proximity labeling assay.

“Proximity labelling assay” was added as suggested by the reviewer.

Thanks!

-Figure 3c and supplementary figure 3b: the log2 ratio is most likely referring to the ratio plus/minus UV?

We apologize if the meaning of the Log2 ratio was not clear. To correct this, the following sentence has been added to the legends of Figure 3c and Supplementary Fig. 4b (former Supplementary Fig 3b): “The data are shown as the Log₂ ratio of protein abundance in each eRIC (or input) sample relative to the average protein abundance in corresponding eRIC (or input) sample”.

Thanks! Important, because many of the other figures employ comparison between +/- UV.

-Supplementary figure 1b and 3c: please catalogue the eRIC specific proteins (with the artificially imputed value) in a separate table.

The information requested by the referee was included in Supplementary Data 1. eRIC specific proteins can be sorted by setting the filter in the “Signal in input” column to FALSE.

Great, thanks!

-Please describe the “Hentze lab” SP3 workflow in more detail, e.g. which organic solvent was employed for bead protein aggregation (acetonitrile, ethanol, propanol)?

As requested by the reviewer, a detailed description of the SP3 workflow used in our study has been added in the methods section (see “Sample preparation for mass spectrometry (MS) and TMT labelling” in page 15).

Perfection. Is there a RNP or protein-RNA crosslinking specific reason to conduct the initial protein aggregation on the magnetic beads at 50% final acetonitrile, since most of the labs use 70 to 80% organic modifier (mostly ACN and ethanol)? Have you checked (having a look at the supernatant) that protein binding is complete?

We thank the reviewer for this perceptive question. As previously shown (Sup. Fig. 1, Moggridge S. et al., *J. Proteome Res.* 2018, 17, 4, 1730–1740; <https://doi.org/10.1021/acs.jproteome.7b00913>), protein binding to beads appears to be similar irrespective of the type (acetonitrile, ethanol, isopropanol, acetone) or concentration (50%, 100%, 150%, 200% relative to the lysate) of the organic solvent. 50% final acetonitrile is the standard condition used in the SP3 protocol in the proteomics core facility at EMBL, not only for RBP-tailored studies but in general. No special considerations in the SP3 pipeline have been introduced to deal with *ex vivo* eRIC samples.

-Instead of harnessing the total proteome, using the list of expressed genes/proteins (in a given organ; should be available from RNAseq data) is usually a much better “background proteome” for GO analyses.

We agree with the reviewer that the list of proteins actually present in the sample in principle represents the most appropriate reference for GO term enrichment analysis. However, using the proteins detected by MS for background correction is not free of bias: some proteins present in very small amounts in input samples may be below the detection limit of MS and be falsely omitted. This was actually the case for 158 (!) proteins that were detected in eRIC eluates but not in the inputs because of the enrichment by oligo(dT) capture. Nonetheless, we performed a new analysis, using the proteome of the input samples as reference. As shown in the tables provided below, we identified GO terms very similar to those of our previous analysis. These include RNA-related clusters such as “translation”, “regulation of RNA stability” and “mRNA splicing, via spliceosome” as well as metabolism-related clusters such as: “tricarboxylic acid cycle”, “fatty acid oxidation” and “NAD binding”. Thus, the method of background correction seems to have little effect on GO term enrichment in this case. Since the core conclusions of either analysis are very similar, we continue to use the original data, which we think are equally robust.

Thanks a lot. I agree with the interpretation of the authors. Was more of an reminder to the proteomics community to make use of the widespread availability of RNAseq data in order to predict expressed proteins.

Reviewer #3 (Remarks to the Author):

This revised manuscript from Perez-Perri and colleagues represents a substantial improvement from the initial submission. I commend the authors for the additional experiments, especially the analysis of “piggy-back” crosslinking, poly(A) vs non-poly(A)

binding, and the expression of new RBPs. The additional experiments and analysis have significantly improved the study, and I support publication in its current form.

We thank this referee for the positive evaluation of our work and for recommending publication in Nature Communications.